# Mitochondrial RNase H1 activity regulates R-loop homeostasis to maintain genome integrity and enable early embryogenesis in *Arabidopsis*

**Lingling Cheng**[1‡], **Wenjie Wang**[1,2‡], **Yao Yao**[1,2‡], **Qianwen Sun**[1,2]*

**1** Center for Plant Biology, School of Life Sciences, Tsinghua University, Beijing, China, **2** Tsinghua-Peking Center for Life Sciences, Beijing, China

‡ These authors contributed equally to this work and are listed alphabetically.
* sunqianwen@mail.tsinghua.edu.cn

**Data Availability Statement:** All relevant data are within the paper and its Supporting Information files.

## Abstract

Plant mitochondrial genomes undergo frequent homologous recombination (HR). Ectopic HR activity is inhibited by the HR surveillance pathway, but the underlying regulatory mechanism is unclear. Here, we show that the mitochondrial RNase H1 AtRNH1B impairs the formation of RNA:DNA hybrids (R-loops) and participates in the HR surveillance pathway in *Arabidopsis thaliana*. AtRNH1B suppresses ectopic HR at intermediate-sized repeats (IRs) and thus maintains mitochondrial DNA (mtDNA) replication. The RNase H1 AtRNH1C is restricted to the chloroplast; however, when cells lack AtRNH1B, transport of chloroplast AtRNH1C into the mitochondria secures HR surveillance, thus ensuring the integrity of the mitochondrial genome and allowing embryogenesis to proceed. HR surveillance is further regulated by the single-stranded DNA-binding protein ORGANELLAR SINGLE-STRANDED DNA BINDING PROTEIN1 (OSB1), which decreases the formation of R-loops. This study uncovers a facultative dual targeting mechanism between organelles and sheds light on the roles of RNase H1 in organellar genome maintenance and embryogenesis.

## Introduction

The R-loop is a 3-stranded nucleic acid structure consisting of an RNA–DNA hybrid strand and a single DNA strand [1–3]. Genome-wide mapping studies showed that R-loops persist throughout the genomes of various species [2,3]. These common chromatin features participate in a number of physiological processes, such as gene expression, DNA replication, and DNA and histone modifications, and DNA damage repair and genome stability [1–3]. Therefore, the formation and resolving of R-loops must be tightly regulated.

Various factors increase the propensity for R-loop formation, such as specific DNA sequences, DNA topology, and regulatory proteins. To maintain balance, many factors restrict R-loop formation, such as ribonucleases, helicases, topoisomerases, degradome components, and RNA binding and processing factors (reviewed in [2,4–6]). Among these factors, the

**Funding:** This work was supported by grants from the Ministry of Science and Technology of China (2016YFA0500800 to Q.S.) and the National Natural Science Foundation of China (grants no. 31822028, 91740105, and 91940306 to Q.S.). The Sun Lab is supported by Tsinghua-Peking Center for Life Sciences, and WW is supported by the postdoctoral fellowship from Tsinghua-Peking Center for Life Sciences. The funders had no role in study design, data collection and analysis, decision to publish, or preparation of the manuscript.

**Competing interests:** The authors have declared that no competing interests exist.

**Abbreviations:** 2D-AGE, two-dimensional agarose gel electrophoresis; aa, amino acids; BSA, bovine serum albumin; CaMV, Cauliflower mosaic virus; ChIP, chromatin immunoprecipitation; CIB, chloroplast isolation buffer; CTS, chloroplast targeting signal; DRIP, DNA:RNA hybrid immunoprecipitation; DSB, double-strand break; GFP, green fluorescent protein; GUS, β-glucuronidase enzyme; HBD, hybrid binding domain; HR, homologous recombination; IR, intermediate-sized repeat; MSH1, *MUTS HOMOLOG1*; mtDNA, mitochondrial DNA; MTS, mitochondrial targeting signal; mtSSB, mitochondrial ssDNA binding protein; OSB1, ORGANELLAR SINGLE-STRANDED DNA BINDING PROTEIN1; POL1B, POLYMERASE 1B; PVDF, polyvinylidene difluoride; qPCR, quantitative PCR; RECA3, RECA HOMOLOG3; RNAi, RNA interference; RT, room temperature; RT-PCR, reverse transcription PCR; SD, standard deviation; SSB, single-stranded DNA-binding protein; WHY2, WHIRLY2.

evolutionarily conserved RNase H proteins (H1 and H2) specifically digest the RNA moiety in RNA:DNA hybrids, thus directly removing R-loops from the genome [7]. Mammalian RNase H1 is dual localized to the nucleus and mitochondria. The loss of function of RNase H1 causes severe diseases in humans, and RNase H1 knockout mutants in mice are embryo lethal [8,9]. RNase H1 is essential for mammalian mitochondrial DNA (mtDNA) replication ([10]; reviewed in [11]). *Arabidopsis thaliana* contains 3 RNase H1s (AtRNH1A, B, and C) that localize to the nuclei, mitochondria, and chloroplasts, respectively [12]. AtRNH1C is important for chloroplast genome stability and plant development [12,13], while the biological functions of the 2 other *Arabidopsis* RNase H1 proteins remain enigmatic.

Mitochondria and chloroplasts are endosymbiotic organelles that play essential roles in plant metabolism, cellular homeostasis, and environmental sensing. The vast majority of mitochondrial and chloroplast proteins are encoded in the nucleus and imported from the cytosol, primarily through thes classical presequence pathway [14,15]. In this pathway, a precursor protein is biosynthesized in the cytosol as a larger preprotein with an N-terminal transit peptide, which directs the protein into the correct organelle/compartment. Mitochondria and chloroplasts contain their own genomes, which encode a limited set of proteins required for their activities. Thus, tight coordination between the nuclear and organellar genomes is important for full functionality.

Compared to the compact circular mtDNA of mammals, which is only 15 to 17 kb in size, plant mitochondrial genomes are much larger, ranging from approximately 200 kb to over 10 Mb [16]. Plant mitochondrial genomes are more complex than their animal counterparts, comprising heterogeneous populations of circular, linear, and multibranched double- and single-stranded molecules [17–19]. This complexity is primarily due to frequent homologous recombination (HR) between repeated sequences, the efficiency of which largely depends on the lengths of the homologous sequences [17,20,21]. Large repeats (>1 kb) undergo high-frequency reciprocal recombination to generate equal isoforms, a process thought to participate in recombination-mediated replication [19,22,23].

By contrast, HR between intermediate-sized repeats (IRs, 50 to 500 bp) is infrequent and asymmetrical, and one of the 2 predicted DNA exchange products preferentially accumulates [17,20,21,23]. Because this recombination activity causes mtDNA rearrangements and genome instability that may affect plant fitness and survival, HR between IRs is rigorously restricted by the HR surveillance pathway. To date, only a few genes involved in this pathway have been identified in *Arabidopsis*, including *MUTS HOMOLOG1* (*MSH1*) [20,24], *ORGANELLAR SINGLE-STRANDED DNA BINDING PROTEIN1* (*OSB1*) [25], *RecA* homolog genes *RECA2* and *RECA3* [26], *RECG1* [27], and SWI/SNF protein complex B (*SWIB5*) [28]. Disrupting these genes increases the ectopic HR frequency between IRs, but the precise regulatory mechanisms are unknown.

Mitochondrial R-loops are important for development and mtDNA replication [11]. Here, to explore the biological functions of R-loop homeostasis in *Arabidopsis* mitochondria, we investigated the roles of the mitochondrial localized RNase H1 protein AtRNH1B. Mutants in *AtRNH1B* did not show any obvious phenotype. However, in a mutant with loss of function of both mitochondrial AtRNH1B and chloroplast AtRNH1C (*atrnh1b/c* double mutant), embryogenesis arrested at the transition stage, leading to embryo lethality. Unexpectedly, AtRNH1C localized to both the mitochondria and chloroplasts in *atrnh1b* mutants and compensated for the function of AtRNH1B in the mitochondria. This dual targeting capacity of AtRNH1C was inhibited in wild-type *Arabidopsis* by an unclear mechanism dependent on a fragment between the transit peptide and hybrid binding domain (HBD) of this protein. We explored the reasons for embryo arrest in the double mutant *atrnh1b/c* and found that it accumulated excessive R-loops and that extensive ectopic HR occurred in its mitochondrial

genome. In addition, the copy number of mtDNA was dramatically reduced in the *atrnh1b/c* mutant. An analysis of the genetic interactions of AtRNH1B/1C with known components of the HR surveillance pathway suggested that the ssDNA binding protein OSB1 inhibits R-loop formation to restrict HR of mtDNA. We propose that a facultative dual targeting mechanism protects mitochondrial RNase H1 to maintain mitochondrial R-loop homeostasis and genome stability, thus safeguarding early embryogenesis in *Arabidopsis*.

## Results

### *Arabidopsis* AtRNH1B is a mitochondrial RNase H1

We previously identified 3 RNase H1 proteins in *Arabidopsis* with different subcellular localizations and characterized the roles of the chloroplast localized protein AtRNH1C in detail [12]. Here, we explored the roles of AtRNH1B, an RNase H1 protein that is targeted to mitochondria. To corroborate the subcellular localization of this protein, we generated stable transgenic *Arabidopsis* plants harboring *AtRNH1B* fused with the green fluorescent protein (*GFP*) gene expressed under its native promoter (*AtRNH1Bpro*:*AtRNH1B-GFP atrnh1b*) and examined GFP signal in both root tips and leaves of the transgenic plants by microscopy. As expected, AtRNH1B-GFP signals consistently colocalized with the magenta signals from Mito-Tracker dye in roots (Fig 1A, upper; S1 Movie). Although the mitochondria in the mesophyll cells from detached leaves could not be efficiently stained by MitoTracker dye, we observed that the GFP signals completely separated from chloroplasts (Fig 1A, bottom). In agreement with these results, a FLAG-tagged AtRNH1B expressed in *atrnh1b* (*AtRNH1Bpro*:*AtRNH1B-FLAG atrnh1b*, hereafter named as *CM-AtRNH1B-FLAG*) was detected in total and mitochondrial fractions but not in chloroplast fractions by immunoblot (Fig 1C).

RNase H1 proteins harbor a conserved catalytic domain (RNase H domain) that specifically digests the RNA moiety in RNA:DNA hybrids [7,29]. To investigate the RNase H activity of AtRNH1B, we purified GST-fused AtRNH1B protein without the mitochondrial targeting signal (MTS), as well as a mutant protein with one amino acid substitution in the conserved site that is crucial for its activity (AtRNH1BM and D191N; S1A and S1B Fig). We incubated the recombinant proteins with RNA:DNA hybrid substrate labeled with FAM (5-carboxyfluorescein). Recombinant AtRNH1B digested the RNA:DNA hybrid substrate as efficiently as commercial RNase H enzyme (Fig 1D, lane 1 to 4; S1C Fig), whereas AtRNH1BM showed no detectable activity (Fig 1D, lane 5 to 6; S1D Fig), indicating that AtRNH1B has canonical RNase H1 activity. Together, these results demonstrate that AtRNH1B is the mitochondrial RNase H1.

### Expression analysis of *AtRNH1B*

To gain insight into the expression of *AtRNH1B*, we generated *AtRNH1Bpro*:*AtRNH1B*-β-glucuronidase enzyme (*GUS*) (expressing *AtRNH1B-GUS* fusion under its own promoter) transgenic plants and performed GUS staining of tissues at different developmental stages. *AtRNH1B* was widely expressed in seedlings, with higher levels in root tips (Fig 2A). In line with Fig 1A, the GUS signal was detected in both epidermal and mesophyll cells (S1E Fig and S2 Movie). Interestingly, upon flowering, *AtRNH1B* was specifically expressed in mature pollen (Fig 2B). Moreover, *AtRNH1B* was expressed throughout embryo development (Fig 2C). These results suggest that AtRNH1B might function in reproductive development and embryogenesis.

### Organellar RNase H1 ensures early embryogenesis

To explore the biological functions of AtRNH1B in *Arabidopsis*, we isolated 2 T-DNA insertional mutants of *AtRNH1B* carrying insertions in the second exon (*atrnh1b-1*) and sixth

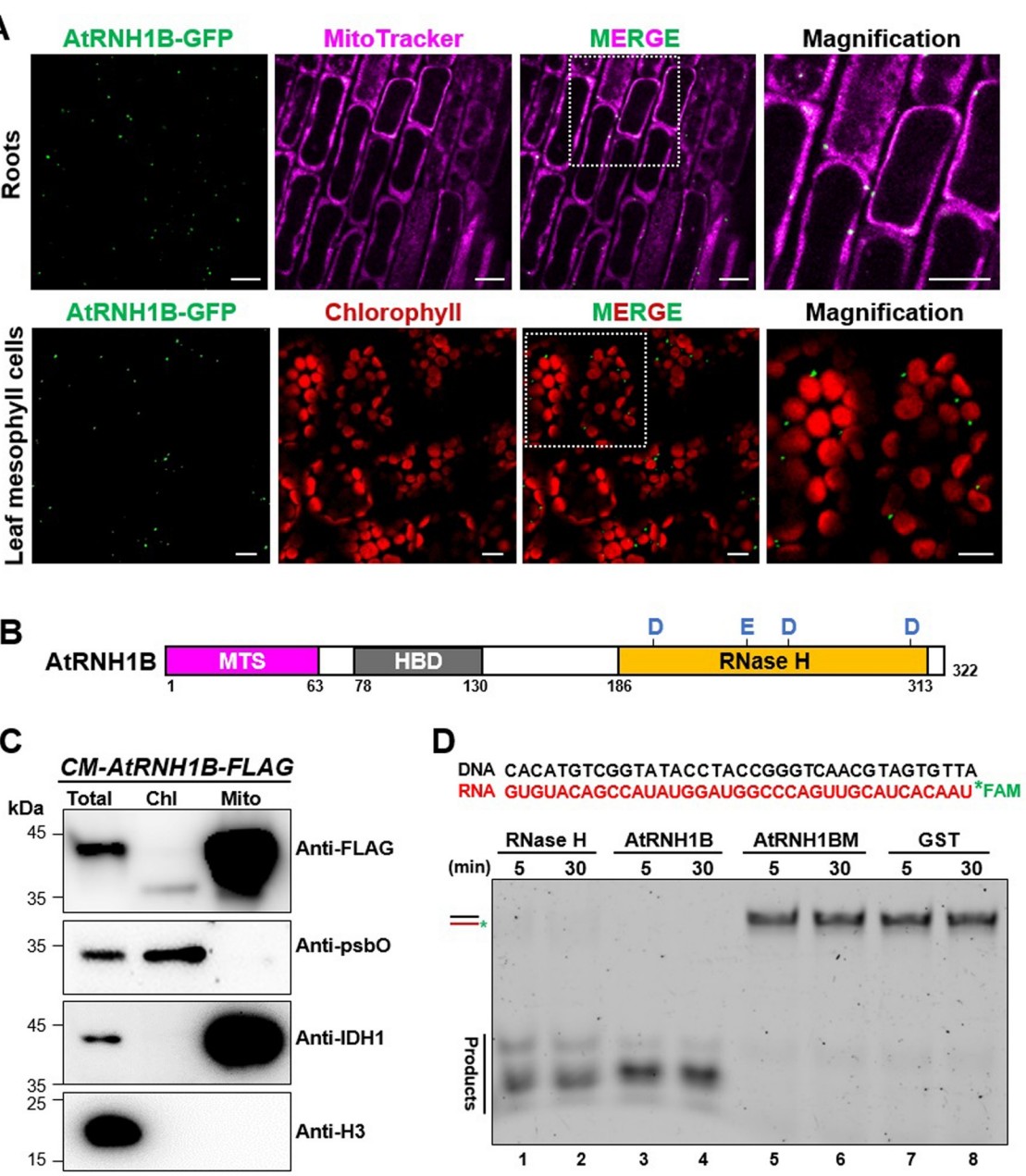

**Fig 1. *Arabidopsis* AtRNH1B is a mitochondrial RNase H1 protein. (A)** Confocal microscopy of roots (upper) and leaves (bottom) of *AtRNH1Bpro:AtRNH1B-GFP atrnh1b* transgenic plants. Green = GFP, magenta = MitoTracker, and red = chlorophyll. White boxes indicate the regions magnified on the right. Scale bars, 10 μm. **(B)** AtRNH1B protein structure. HBD, hybrid binding domain; MTS, mitochondrial targeting signal; RNase H, catalytic domain (containing 4 conserved active sites: D191, E231, D255, and D305). D, Asp; E, Glu. **(C)** Characterize the localization of AtRNH1B protein by immunoblot. Intact chloroplasts and mitochondria were isolated from 3-week-old *CM-AtRNH1B-FLAG* transgenic plants. Anti-FLAG monoclonal antibody was used to detect the AtRNH1B-FLAG, and polyclonal antibodies anti-psbO, anti-IDH1, and anti-H3 were used to indicate chloroplast, mitochondria, and total protein fractions, respectively. Chl, proteins from isolated chloroplasts; Mito, proteins from isolated mitochondria; Total, total proteins from leaves. **(D)** RNase H activity of AtRNH1B and AtRNH1BM. The RNA:DNA hybrid substrate with FAM-labeled RNA (100 nM) was incubated with commercial RNase H and 0.55 μg purified GST-AtRNH1B, 0.55 μg GST-AtRNH1BM, and 2.8 μg GST protein for 5 minutes and 30 minutes. The data underlying this figure can be found in S1 Raw Images. GFP, green fluorescent protein; GST, glutathione-S-transferase; H3, histone 3; IDH1, isocitrate dehydrogenase 1; psbO, photosystem II subunit O.

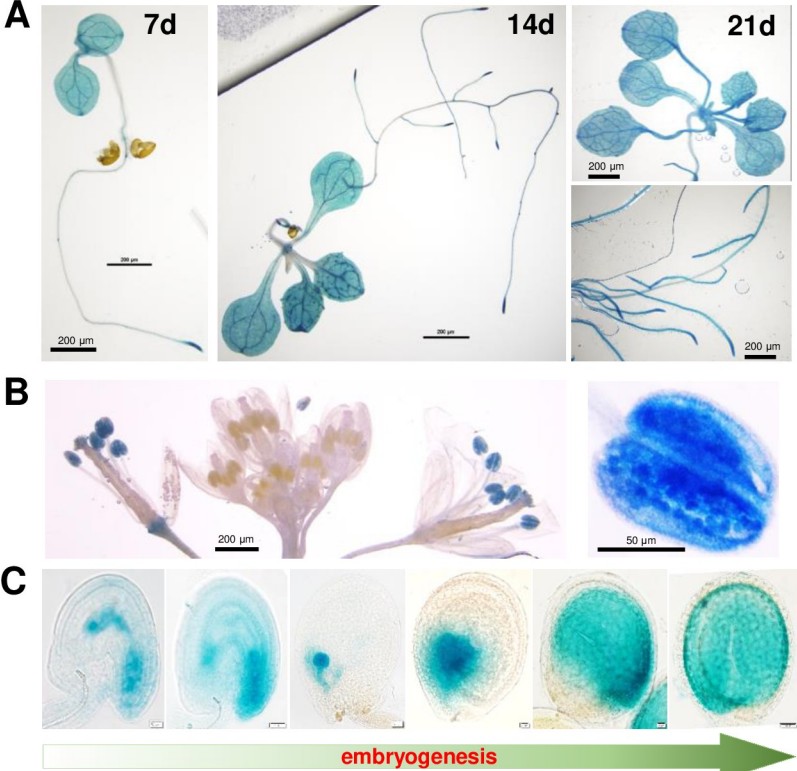

**Fig 2. Expression analysis of AtRNH1B in *Arabidopsis*.** GUS staining shows the expression pattern of *AtRNH1B* in *AtRNH1Bpro*:*AtRNH1B-GUS atrnh1b-1* transgenic plants in different tissues and developmental stages. **(A)** 7-, 14-, and 21-day-old seedlings. Scale bars, 200 μm. **(B)** Open flowers and buds (left) and pollen (right). Scale bars are indicated. **(C)** Seeds at different developmental stages (unfertilized ovule, fertilized ovule, globular, heart, torpedo, and mature). Scale bars, 20 μm. GUS, β-glucuronidase enzyme.

intron (*atrnh1b-2*), respectively (S2A Fig). Reverse transcription PCR (RT-PCR) analysis showed that the expression of *AtRNH1B* was abolished in the mutants (S2B and S2C Fig). However, the phenotypes of both *atrnh1b-1* and *atrnh1b-2* were similar to those of wild-type Col-0 (S2D Fig), suggesting that RNase H1 proteins, especially organellar proteins, might be functionally redundant. To test this hypothesis, we tried to combine *atrnh1b-1* with *atrnh1c* by crossing. After several attempts, we failed to obtain a homozygous *atrnh1b-1 atrnh1c* double mutant (hereafter referred to as *atrnh1b/c*). Instead, we produced the *atrnh1b-1 atrnh1c$^{+/-}$* double mutant (hereafter referred to as *atrnh1b 1c$^{+/-}$*), which is homozygous for *atrnh1b* and heterozygous for *atrnh1c*. Approximately 25% of the seeds in *atrnh1b 1c$^{+/-}$* siliques were abnormal (Fig 3A and 3B), while the seeds in *atrnh1b* and *atrnh1c* were completely normal (S3A Fig). When we germinated the seeds on medium, they segregated at a ratio of 2 *atrnh1b 1c$^{+/-}$* to 1 *atrnh1b* (S3B Fig). Reciprocal cross between Col-0 and *atrnh1b 1c$^{+/-}$* showed that both the *atrnh1b 1c$^{+/-}$* male and female gametophytes are fertile (S3A Fig). These results indicate that the simultaneous knockout of *AtRNH1B* and *AtRNH1C* leads to seed abortion.

To further confirm that the knockout of *AtRNH1B* was responsible for seed abortion in *atrnh1b 1c$^{+/-}$* plants, we performed complementation analysis using genomic constructs expressing AtRNH1B (*AtRNH1Bpro*:*AtRNH1B-GFP*) and AtRNH1B without the MTS (*AtRNH1Bpro*:*AtRNH1BΔMTS-GFP*). The seed abortion was rescued in *AtRNH1B* transformants, but not in *AtRNH1BΔMTS* transformants (Fig 3A and 3B). In addition, we generated the *atrnh1b/c* double mutant carrying the *AtRNH1B* transgene, which showed the same

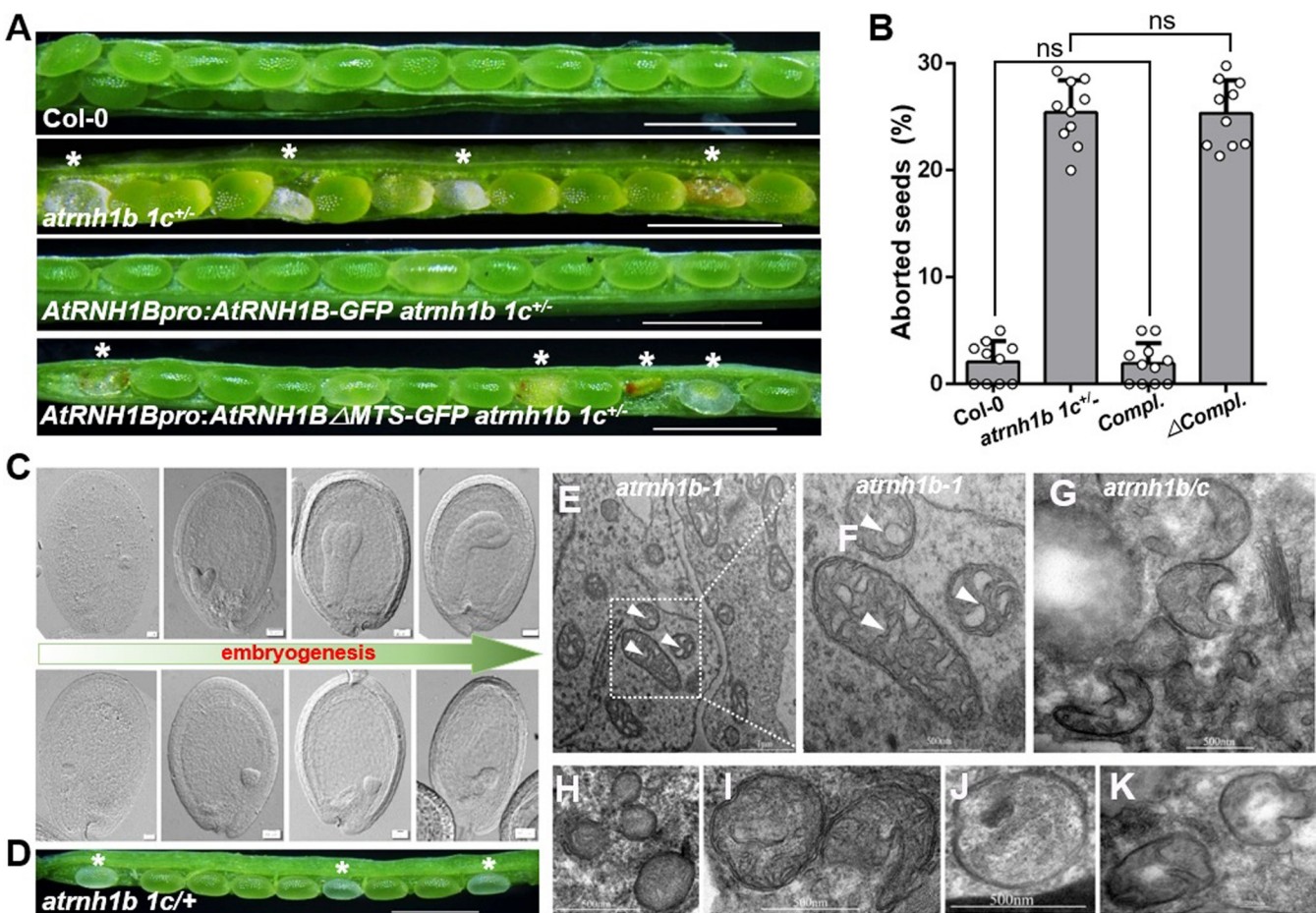

**Fig 3. Organellar RNase H1 ensures successful early embryogenesis. (A)** Dissected siliques of wild-type Col-0 and *atrnh1b 1c$^{+/−}$* plants and *atrnh1b 1c$^{+/−}$* plants complemented with *AtRNH1Bpro:AtRNH1B-GFP* or *AtRNH1Bpro:AtRNH1BΔMTS-GFP* (*AtRNH1B* lacking the MTS). Asterisks indicate lethal seeds. Scale bars, 200 μm. **(B)** Statistical analysis of aborted seeds in the siliques shown in (A). *compl.* and *Δcompl.* refer to *AtRNH1Bpro:AtRNH1B-GFP atrnh1b 1c$^{+/−}$* and *AtRNH1Bpro:AtRNH1BΔMTS-GFP atrnh1b 1c$^{+/−}$*, respectively. Data are from 10 repeats (silique examinations) and 5 siliques were examined each time. The total aborted seeds/population sizes of the 4 genotypes are 13/513, 382/1,458, 8/482 and 408/1,628 seeds, respectively. Data are mean values ± SD; circles show the original data. Significance test was performed using 1-way ANOVA, and ns indicates no significance. **(C)** Seed clearing to observe normal (upper) and abnormal (lower) embryos from siliques of *atrnh1b 1c$^{+/−}$* plants. The embryos are in the globular, heart, torpedo, and mature stages, respectively. Scale bars, 20 μm. **(D)** Dissected silique of *atrnh1b 1c/+* in heart stage. Asterisks indicate white abnormal seeds. Scale bars, 200 μm. **(E–K)** Transmission electron microscopy of seeds from *atrnh1b-1* plants (E–F) and abnormal seeds from *atrnh1b 1c$^{+/−}$* plants (G–K). White arrowheads indicate internal cristae membranes. The data underlying this figure can be found in S1 Data. ANOVA, analysis of variance; MTS, mitochondrial targeting signal; SD, standard deviation.

yellowish phenotype as *atrnh1c* (S3C Fig). However, no *atrnh1b/c AtRNH1BΔMTS* transgenic plants were identified (S3C Fig).

To investigate the defects in homozygous *atrnh1b/c* embryos at the cytological level, we examined ovule development in *atrnh1b 1c$^{+/−}$* using a whole mount clearing technique. We analyzed seeds at different stages of development from plants after self-pollination. We could not distinguish abnormal embryos from normal embryos before the heart stage (Fig 3C). However, after the transition from the globular to heart stage, *atrnh1b 1c$^{+/−}$* contained both green ovules with heart stage embryos and white ovules with embryos arrested at the transition stage (Fig 3C and 3D). During later development, the embryos in green ovules continued to undergo embryogenesis, and the ovules developed into mature seeds, whereas the embryos in white ovules were trapped at the transition stage, and the ovule ultimately shriveled (Figs 3A, 3C and 3D and S3D).

As lack of the mitochondrial localized RNase H is responsible for embryo lethality, we wondered whether mitochondrial morphology was affected in *atrnh1b/c* embryos. We examined the ultrastructure of the mitochondria by transmission electron microscopy. As shown in Fig 3D–3J, the mitochondrial cristae were fragmented or largely absent in *atrnh1b/c* embryos compared to embryos from Col-0 and the single mutants (Figs 3D–3J and S3E). Moreover, the number of mitochondria in seeds of *atrnh1b/c* was reduced (S3F Fig). Collectively, these results suggest that disrupting both AtRNH1B and AtRNH1C impeded mitochondrial function during early embryogenesis, thereby causing *atrnh1b/c* embryos to arrest at the transition stage and leading to seed abortion.

## Dual localization of AtRNH1C in chloroplasts and mitochondria

The finding that *atrnh1b* showed a wild-type phenotype but the homozygous *atrnh1b/c* double mutants were embryo lethal suggested that AtRNH1C and AtRNH1B function redundantly in mitochondria, raising the intriguing question of how the chloroplast localized AtRNH1C compensates for the absence of AtRNH1B in mitochondria. We previously demonstrated that AtRNH1C localizes to the chloroplast in wild-type Col-0 plants [12]. Thus, we hypothesized that AtRNH1C is translocated into both the chloroplasts and mitochondria in the absence of AtRNH1B.

To test this possibility, we reanalyzed the subcellular localization of AtRNH1C in Col-0 and *atrnh1b* plants. Specifically, we expressed the full-length coding sequence of *AtRNH1C* driven by the Cauliflower mosaic virus (CaMV) 35S promoter in mesophyll protoplasts from *Arabidopsis* plants of different genotypes. In agreement with our previous data, the fusion protein was predominantly transported into chloroplasts in the Col-0 background (Fig 4A). However, in *atrnh1b*, GFP fluorescence was detected in both the chloroplasts and mitochondria (Fig 4A). As 35S promoter may lead to mistargeting especially for subcellular localized proteins, we then examined the localization of AtRNH1C using stable transgenic plants containing *AtRNH1Cpro:AtRNH1C-GFP*. In *AtRNH1Cpro:AtRNH1C-GFP atrnh1c*, the yellowish phenotype of *atrnh1c* was rescued, but the GFP signals in leaf tissues was too low to be captured by microscopy (Fig 4B). We had previously reported that, although the AtRNH1C protein in chloroplast can be detected by western blot, the expression of *AtRNH1C* was very weak [12,13]. Strikingly, the GFP signals were clearly observed in *AtRNH1Cpro:AtRNH1C-GFP atrnh1b*, and, moreover, the signals were mainly localized into mitochondria (Fig 4B). Similar results were obtained in root tissues (S4A Fig). Then, we further determined the subcellular localization of AtRNH1C-GFP in the stable transgenic plants through immunogold electron microscopy. Although the green fluorescence of AtRNH1C-GFP was not detected in *AtRNH1Cpro:AtRNH1C-GFP atrnh1c*, the gold particles recognizing GFP were observed in chloroplasts but not in mitochondria (Fig 4C). In *AtRNH1Cpro:AtRNH1C-GFP atrnh1b*, the immunogold labeled GFP signals were detected in both chloroplast and mitochondria (Fig 4C). Taken together, these results indicate that, in the absence of AtRNH1B, partial AtRNH1C can be transported into mitochondria.

Next, we explored the mechanisms that restrict the access of AtRNH1C to mitochondria under normal conditions. We evaluated the organellar import specificity of the in silico predicted chloroplast targeting signal (CTS) of AtRNH1C (hereafter referred to as $CTS^{1C}$) in transiently transformed protoplasts (S4B Fig). In contrast to the chloroplast localization of full-length AtRNH1C fused with GFP ($CTS^{1C}$-AtRNH1C-GFP) (Fig 5A and 5D), $CTS^{1C}$ alone delivered the carboxyl-terminal GFP moiety ($CTS^{1C}$-GFP) into both mitochondria and chloroplasts in wild-type protoplasts (Fig 5A and 5B), whereas AtRNH1C without the CTS (AtRNH1CΔCTS-GFP) did not show any organellar specificity (Fig 5A and 5C). These observations suggest that signals in AtRNH1C inhibit the mitochondrial localization of this protein.

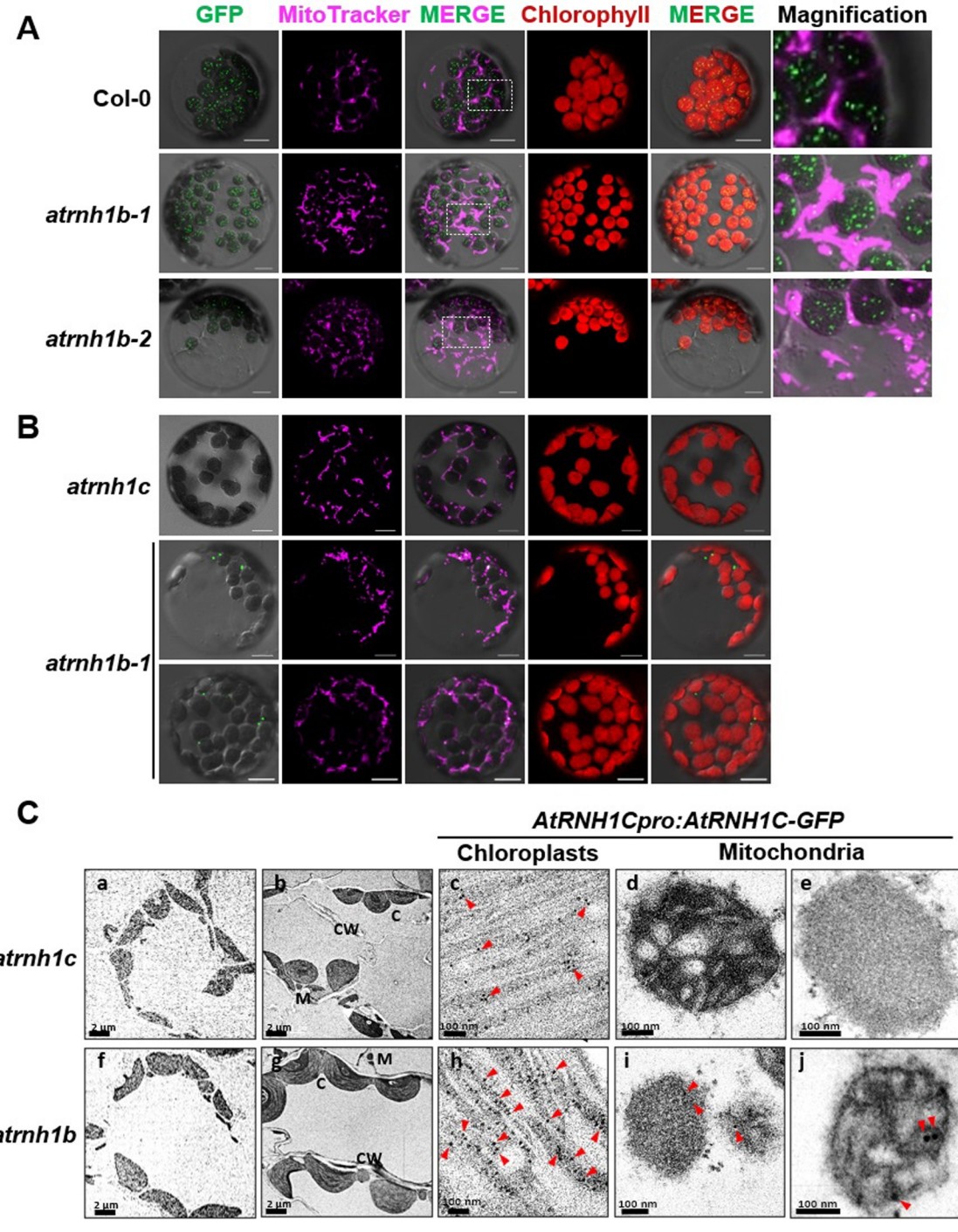

**Fig 4. AtRNH1C can localize to mitochondria in the absence of AtRNH1B. (A)** Col-0, *atrnh1b-1*, and *atrnh1b-2* protoplasts were transformed with *AtRNH1C-GFP* driven under 35S promoter. Green = GFP, magenta = MitoTracker, and red = chlorophyll. White boxes indicate the regions magnified on the right. Scale bars, 10 μm. **(B)** Protoplasts from the leaves of *AtRNH1Cpro*: *AtRNH1C-GFP* transgenic plants in the *atrnh1c* and *atrnh1b-1* backgrounds. **(C)** Immunogold labeling detects dual localization event of AtRNH1C in the absence of AtRNH1B. (a and f) Preimmune controls showed no gold labeling in the cells. (b–e) AtRNH1C-GFP signal appeared to be clustered in chloroplast exclusively in *AtRNH1Cpro:AtRNH1C-GFP atrnh1c* transgenic plants. (g–j) AtRNH1C-GFP signal can be detected in both chloroplast and mitochondria in *AtRNH1Cpro:AtRNH1C-GFP atrnh1b* transgenic plants. C, chloroplast; CW, cell walll; GFP, green fluorescent protein; M, mitochondrion.

Surprisingly, AtRNH1C converted the mitochondrial-targeting signal of AtRNH1B (MTS[1B]) into the CTS, as MTS[1B]-AtRNH1C-GFP predominantly localized to chloroplasts, whereas MTS[1B] was specifically imported GFP into mitochondria (Fig 5A, 5E, and 5F), reinforcing the notion that the mitochondrial specificity of AtRNH1C was inhibited.

To identify the crucial signals responsible for inhibiting the mitochondrial localization of AtRNH1C in wild-type Col-0, we compared the amino acid sequences of AtRNH1C and AtRNH1B. These 2 sequences are highly similar, except for a variable region located between the transit peptides and HBD and an extra fragment in AtRNH1C (S4B Fig). After deleting amino acids (aa) 69 to 77, but not 163 to 183 aa, in AtRNH1C, MTS[1B] successfully imported AtRNH1C into mitochondria (Fig 5G and 5H). Moreover, results from the stable transgenic plants also showed that, under a dual localization capacity of CTS signal (but not MTS), loss of 69 to 77 aa confers the dual localization of AtRNH1C (Fig 5I). These results indicate that the 69 to 77 aa sequence of AtRNH1C is essential to inhibit an ability of this protein localizes into mitochondria.

## Depletion of *AtRNH1B* promotes *AtRNH1C* expression

In addition to the altered subcellular localization of AtRNH1C, AtRNH1C-GFP produced much stronger fluorescent signals in *atrnh1b* than in *atrnh1c* (Fig 4B), mainly in mitochondria, suggesting that *AtRNH1C* expression increased in the absence of AtRNH1B. To explore whether the expression of *AtRNH1C* changed when *AtRNH1B* was depleted, we performed RT-PCR, finding that *AtRNH1C* was expressed at significantly higher levels in *atrnh1b* compared to Col-0 (Fig 6A). We then generated *AtRNH1Cpro:AtRNH1C-GUS* (expressing *AtRNH1C-GUS* fusion under its own promoter) transgenic plants in the *atrnh1b-1* and *atrnh1c* mutant backgrounds, respectively. GUS staining revealed very little *AtRNH1C-GUS* reporter activity in *atrnh1c* but extremely strong *AtRNH1B-GUS* reporter activity in *atrnh1b*. Consistent with our RT-PCR results, the expression of AtRNH1C was greatly enhanced in *atrnh1b* compared to *atrnh1c* (Figs 6B and S4C). Since AtRNH1B is specifically expressed in embryos (Fig 2C), we also observed significantly increased AtRNH1C-GUS expression in *atrnh1b-1* versus *atrnh1c* embryos (Fig 6C). The increase of AtRNH1C-GUS in *atrnh1b* was confirmed through immunoblot analysis (Fig 6D). Consistent with previous observation (Fig 4B and 4C; there are much more immunogold labeled GFP particles detected in *atrnh1b* chloroplast than that in *atrnh1c* chloroplast), the accumulation of AtRNH1C in the mitochondria of *atrnh1b* was remarkable (Fig 6D). Moreover, the level of AtRNH1C was also slightly increased (Fig 6D). These results further support the notion that AtRNH1C makes up for the loss of AtRNH1B in the *atrnh1b* mutant potentially to compensate the mitochondrial RNase H1 function.

Based on above results, we propose that a facultative dual targeting mechanism could ensure the mitochondria having RNase H1 function (Fig 6D). In the presence of functional AtRNH1B, the mitochondrial localization of AtRNH1C driven by the CTS is inhibited by another signal that prevents the overaccumulation of RNase H1 in mitochondria via an unknown mechanism. In cells lacking mitochondrial localized RNase H1 (e.g., when AtRNH1B is depleted, as in the current study), the inhibition of the mitochondrial localization of AtRNH1C is relieved and its expression increases, allowing sufficient amounts of AtRNH1C to be delivered into the mitochondria to ensure the proper functioning of mitochondrial RNase H1. This facultative dual targeting protective mechanism might ensure that plants properly respond to environmental stimuli.

## Knocking down *AtRNH1B* in *atrnh1c* causes sterility

As *atrnh1b/c* homozygous plants could not be generated for further functional analysis of plant mitochondrial RNase H1, we produced *AtRNH1B* knockdown transgenic plants in the

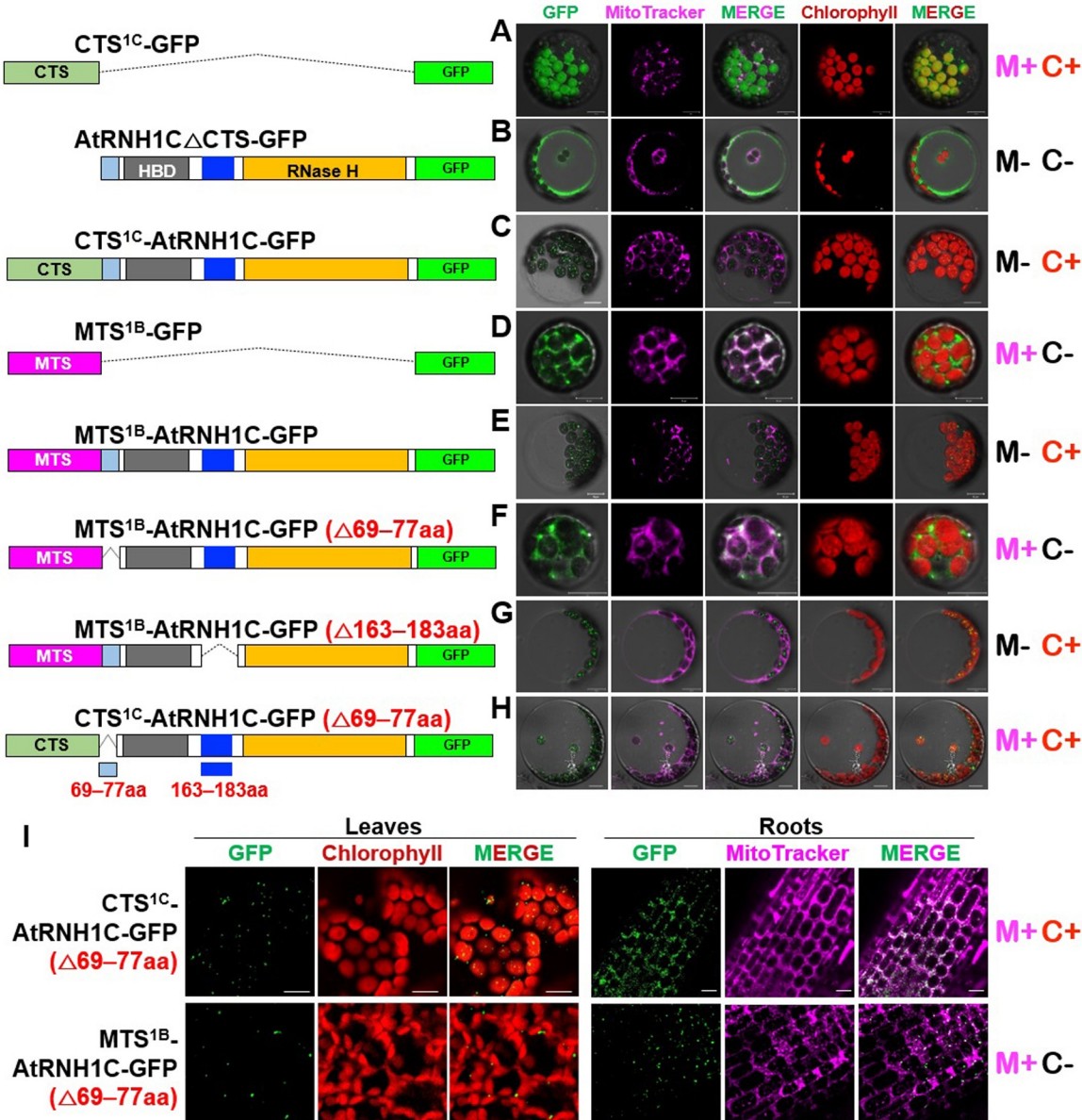

**Fig 5. The mitochondrial localization of AtRNH1C is inhibited by aa 69–77 of this protein.** Schematic drawing the protein structures for expression in (A–H) were shown in the left. **(A–H)** Col-0 protoplasts were transformed with different vectors expressing proteins. **(I)** Subcellular localization of GFP-fusion proteins of CTS$^{1C}$-AtRNH1C (△69–77aa), MTS$^{1B}$-AtRNH1C (△69–77aa) in Col-0 transgenic plants. Green = GFP, magenta = MitoTracker, and red = chlorophyll. Scale bars, 10 μm. The GFP and MitoTracker images were merged to show the colocalization of the GFP signal and mitochondria. The summary of the results was listed in the right. "M+" indicates mitochondrial localization; "M−" indicates no mitochondrial localization; "C+" indicates chloroplast localization; and "C−" indicates no chloroplast localization. 1C, AtRNH1C; 1B, AtRNH1B; aa, amino acid; CTS, chloroplast targeting signal; HBD, hybrid binding domain; MTS, mitochondrial targeting signal.

*atrnh1c* mutant background by RNA interference (RNAi) (S5A Fig, hereafter referred to as *AtRNH1B$^{RNAi}$ atrnh1c*). Although the resulting individual transformants expressed *AtRNH1B* at different levels in *atrnh1c* (S5B Fig), all these plants displayed a yellowish phenotype like that of *atrnh1c* plants at the vegetative stage (S5C Fig), indicating that mitochondrial AtRNH1B is not involved in chloroplast development.

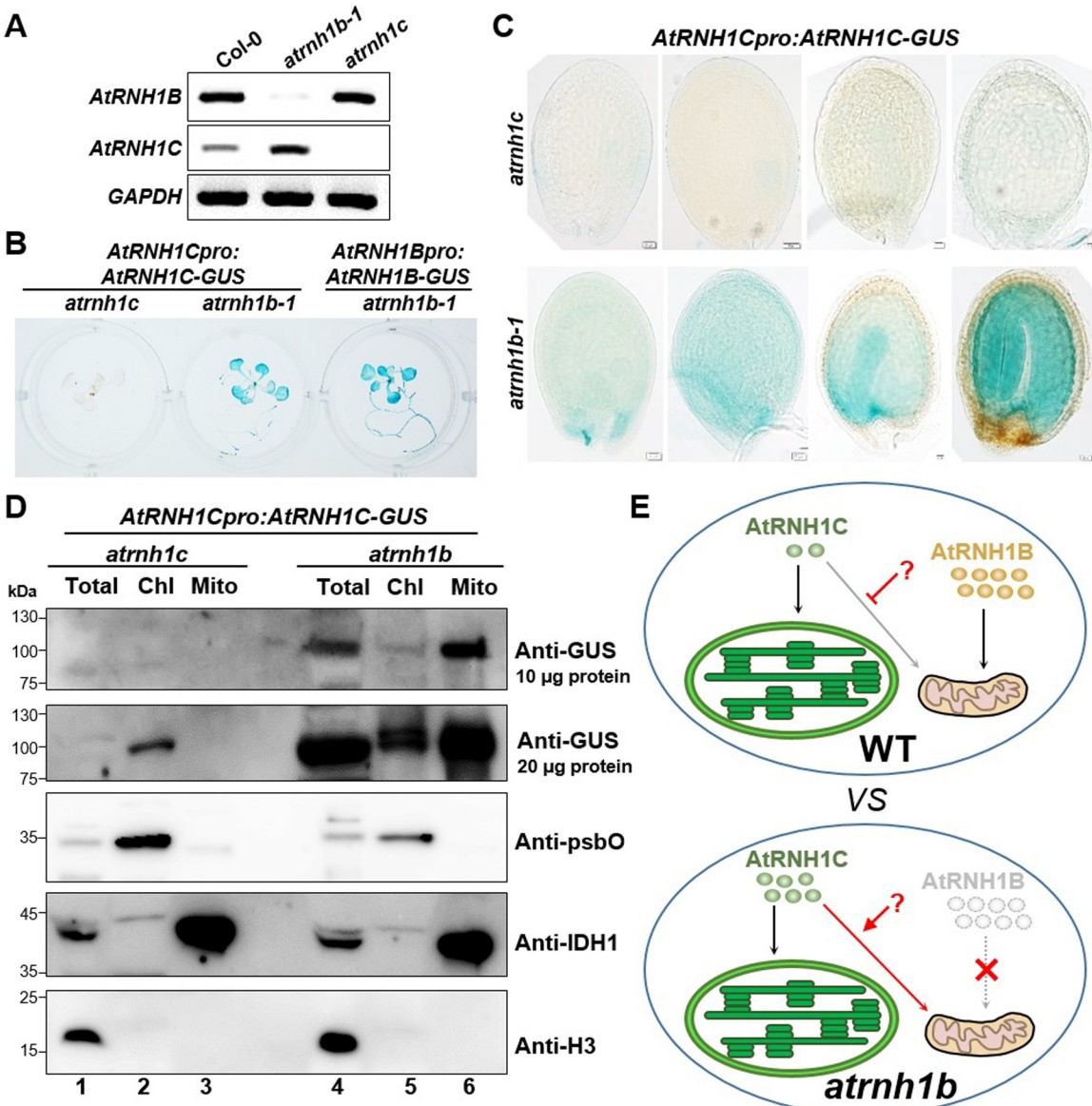

**Fig 6. The expression of AtRNH1C increases in *atrnh1b*.** (A) A total of 28 RT-PCR cycles were used to detect the expression of *AtRNH1B* and *AtRNH1C* in Col-0, *atrnh1b-1*, and *atrnh1c*. *GAPDH* was used as the reference gene. (B, C) GUS staining of 2-week-old seedlings (B) and seeds at the globular, heart, torpedo, and mature embryo stages (C) from plants expressing *AtRNH1Cpro*: *AtRNH1C-GUS* in the *atrnh1c* and *atrnh1b-1* backgrounds. (D) Characterize the localization of AtRNH1C protein by immunoblot. Intact chloroplasts and mitochondria were isolated from 3-week-old *AtRNH1Cpro:AtRNH1C-GUS atrnh1c* and *AtRNH1Cpro*: *AtRNH1C-GUS atrnh1b* transgenic plants. Anti-GUS polyclonal antibody was used to detect the AtRNH1C-GUS, and polyclonal antibodies anti-psbO, anti-IDH1, and anti-H3 were used to indicate chloroplast, mitochondria, and total protein fractions, respectively. Chl, proteins from isolated chloroplasts; Mito, proteins from isolated mitochondria; Total, total proteins from leaves. (E) A facultative dual targeting mechanism protects mitochondrial RNase H1. In the wild-type, AtRNH1C and AtRNH1B are predominately transported to the chloroplast and mitochondria, respectively, and the localization of AtRNH1C to the mitochondria is self-inhibited by an unknown mechanism. The level of AtRN1HB is much higher than that of AtRNH1C. When *AtRNH1B* is mutated (*atrnh1b*), AtRNH1C recovers its dual localization to both the chloroplast and mitochondria and its expression level increases, thus safeguarding mitochondrial function. The data underlying this figure can be found in S1 Raw Images. GUS, β-glucuronidase enzyme; H3, histone 3; IDH1, isocitrate dehydrogenase 1; psbO, photosystem II subunit O; RT-PCR, reverse transcription PCR; WT, wild type.

The 3 most effective RNAi lines (*AtRNH1B^RNAi atrnh1c*-#1, *atrnh1c*-#2, and *atrnh1c*-#3) with dramatically reduced *AtRNH1B* expression showed sterility (S5B, S5D and S5E Fig). Transmission electron microscopy of young siliques from these lines showed that the

mitochondrial morphology was abnormal in sterile plants, in contrast to *atrnh1c* (S5F Fig), and was similar to that observed in *atrnh1b/c* (Fig 3D). Pollen viability assay and reciprocal cross indicated that both the male and female gametophytes are fertile in *AtRNH1B^RNAi* *atrnh1c* (S5G and S5H Fig). These results confirm the notion that mitochondrial RNase H1 is important for reproduction.

## Mitochondrial RNase H1 helps maintain R-loops and functions in HR of mtDNA

Compared to the plastid genome, the plant mitochondrial genome is rich in the repeated sequences that can mediate ectopic HR [30], but the underlying mechanism remains unclear. R-loops promote HR for DNA repair [31,32]. Therefore, we wondered whether mitochondrial RNase H1 is involved in regulating HR at those repeated mtDNA sequences (S6A Fig). We performed DNA:RNA hybrid immunoprecipitation (DRIP)-quantitative PCR (qPCR) to investigate the effects of RNase H1 on R-loops in the repeats of mitochondrial genome. Since we determined that mitochondrial RNase H1 plays important roles in seed development, but we were unable to purify enough mtDNA from *atrnh1b/c* seeds for DRIP-qPCR, we used siliques from Col-0, *atrnh1c*, and *AtRNH1B^RNAi* *atrnh1c* (#1, #2) plants for DRIP-qPCR. We analyzed the *26S* gene and 5 IRs that drive HR in mtDNA (EE1, EE2, L, T1, and T2, S6A Fig), and the chloroplast *23S* gene was included as a positive control. Col-0 and *atrnh1c* accumulated similar levels of mitochondrial R-loops, whereas *AtRNH1B^RNAi* *atrnh1c* transgenic plants produced significantly more mitochondrial R-loops at the regions examined (Fig 7A). These results further demonstrate the RNase H activity of endogenous RNase H1 (Fig 1).

We then studied the effects of greater numbers of R-loops on HR frequency in mtDNA. The accumulation of crossover products resulting from IR-mediated recombination can be measured by qPCR and used as a measure of HR frequency [26]. Changes in the accumulation of crossover products and mtDNA copy number have been observed in mutants with altered mitochondrial HR [26,27]. Here, we examined the crossover products in the seeds of mutant and wild-type plants. We selected several pairs of repeats that are undergo active recombination and are amenable to relative quantification of the corresponding crossover products by qPCR (S6A Fig, as annotated in the recently published mtDNA sequence of ecotype Columbia (Col-0); accession no. BK010421 [33]) as markers for qPCR quantification of mtDNA recombination. We used different pairs of primers designed to specifically amplify a selection of repeats and their corresponding crossover products for qPCR (Fig 7B) [26,27]. We observed a dramatic increase in the amount of one crossover product of repeats L, I, and T (sequences L2/1, I1/2 and T2/1, respectively) in the *atrnh1b/c* double mutant compared to Col-0 and the single mutants (Fig 7C), indicating that the absence of mitochondrial RNase H1 triggered the asymmetrical recombination at these IRs. In the case of repeats EE, K, and X, both crossover products accumulated in *atrnh1b/c* (Fig 7C). These results suggested that R-loops might promote HR in *Arabidopsis* mitochondrial genome.

Because frequent ectopic HR is a source of genome instability, therefore, we determined the relative copy numbers of several mtDNA genes and the repeats. We quantified the copy numbers of 5 gene sequences and 6 repeats pairs in the mitochondrial genome and standardized the results against nuclear DNA. The *atrnh1b/c* mutant had dramatically reduced mtDNA copy numbers compared to the wild-type Col-0 and single mutants (Fig 7D and 7E), suggesting that the high-frequency HR triggered by IRs reduces mtDNA copy number. In addition, we examined the HR frequency of the repeats L and EE and the relative copy numbers of *18S* and *COX2* in various tissues at different developmental stages. The changes in mtDNA copy number contrasted with the changes in HR frequency. However, similar to the expression

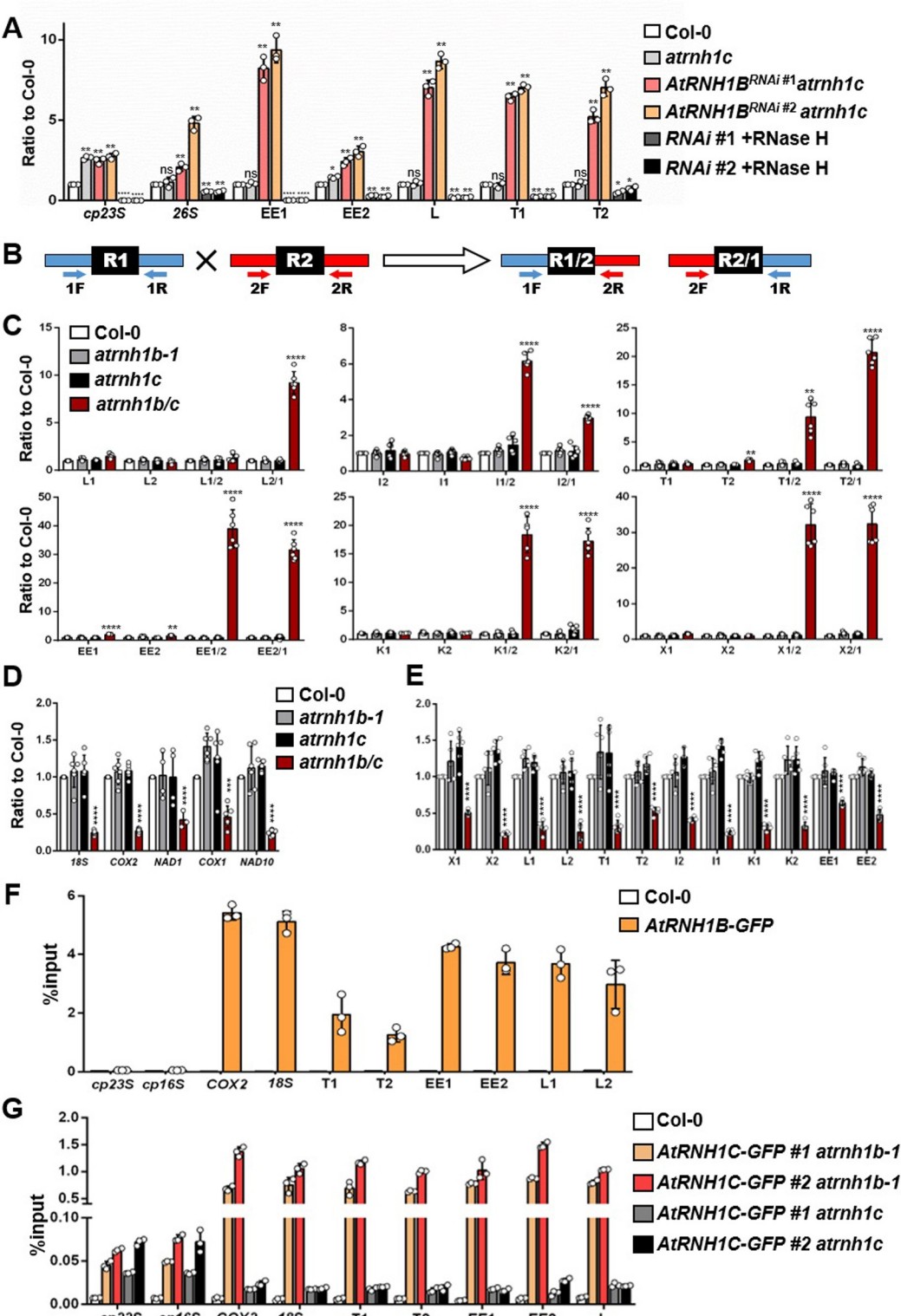

**Fig 7. *Atrnh1b/c* exhibits increased R-loop formation and HR, leading to reduced mtDNA levels. (A)** DRIP-qPCR of siliques from Col-0, *atrnh1c*, and *AtRNH1B^RNAi^ atrnh1c* lines (*RNAi #1* and *#2*), including the chloroplast gene *23S* (*cp23S*) and mitochondrial gene *26S* and repeats EE1, EE2, L, and T. Data are normalized to Col-0 and shown as mean values ± SD; circles show the original data from 3 replicates. The *cp23S* locus was used as a positive control. *RNAi #1/ #2* +RNase H1 were used as negative controls. One-way ANOVA compared to Col-0 for each primer. ns, no significance; *, $p < 0.05$; **,

$p < 0.01$; ***, $p < 0.001$; ****, $p < 0.0001$. **(B)** Simplified scheme of qPCR amplification of sequence R1 and R2 comprising a repeated sequence (black R box) and the crossover products R1/2 and R2/1. Scheme adapted from [17]. **(C)** qPCR of the parental sequences and crossover products (as depicted in B) of repeats L, I, T, EE, K, and X in seeds at the globular stage from Col-0, *atrnh1b-1*, *atrnh1c*, and *atrnh1b/c*. Mitochondrial genes *18S* and *COX2* were used as reference genes. One-way ANOVA compared to Col-0 for each primer. **, $p < 0.01$; ****, $p < 0.0001$. **(D, E)** Relative quantification of copy numbers of mtDNA gene sequences (*18S*, *COX2*, *NAD1*, *COX1*, and *NAD10*) (D) and repeated sequences (E) in seeds at the globular stage from Col-0, *atrnh1b-1*, *atrnh1c*, and *atrnh1b/c*. Nuclear genes *UBC* and *ACT2* were used as reference genes. One-way ANOVA compared to Col-0 for each primer. ***, $p < 0.001$; ****, $p < 0.0001$. **(F)** AtRNH1B ChIP-qPCR using *AtRNH1Bpro:AtRNH1B-GFP atrnh1b/c* (shown as *AtRNH1B-GFP* in brief) transgenic plants and Col-0. The *cp23S* and *cp16S* loci were used as negative controls. **(G)** AtRNH1C ChIP-qPCR using *AtRNH1Cpro:AtRNH1C-GFP* (shown as *AtRNH1C-GFP* for briefness) in *atrnh1b-1* and *atrnh1c*. Data are normalized as %input and shown as mean values ± SD; circles show the original data of 3 replicates. The *cp23S* and *cp16S* loci were used as positive controls. The data underlying this figure can be found in S1 Data. *ACT2*, *ACTIN 2*; ANOVA, analysis of variance; ChIP, chromatin immunoprecipitation; *COX1*, *CYTOCHROME OXIDASE 1*; *COX2*, *CYTOCHROME OXIDASE 2*; DRIP, DNA:RNA hybrid immunoprecipitation; HR, homologous recombination; mtDNA, mitochondrial DNA; *NAD1*, *NADH DEHYDROGENASE 1*; *NAD10*, *NADH DEHYDROGENASE 10*; qPCR, quantitative PCR; R1, repeat 1; R2, repeat 2; RNAi, RNA interference; SD, standard deviation; *UBC*, *UBIQUITIN-CONJUGATING ENZYME*.

pattern of AtRNH1B, the relative copy numbers of *18S* and *COX2* and the crossover products were highest in roots, flowers, and young siliques (Figs 2 and S6B and S6C).

We further investigated the association of AtRNH1B with mtDNA by performing a modified chromatin immunoprecipitation (ChIP) experiment using transgenic plants carrying GFP-tagged AtRNH1B (Figs 7F and S1A; see details in Methods). ChIP-qPCR showed that AtRNH1B clearly bound to the repeated regions and coding genes of mtDNA, whereas no clear interaction with chloroplast DNA was detected (*16S* and *23S*; Fig 7F). Because AtRNH1C was translocated into the mitochondria to take over the function of AtRNH1B in the *atrnh1b* mutant (Fig 4), we also compared the binding activity of AtRNH1C with mtDNA in *AtRNH1Cpro:AtRNH1C-GFP atrnh1c* and *AtRNH1Cpro:AtRNH1C-GFP atrnh1b* transgenic plants. In line with the subcellular localization observed (Figs 4 and 5), the binding of AtRNH1C with mtDNA was very close to negative control in *atrnh1c* complemental lines, while loss of AtRNH1B dramatically triggered the association of AtRNH1C with mtDNA (Fig 7G). These results also stress the idea that the mitochondrial localization of AtRNH1C is inhibited in normal condition, but activated in absence of AtRNH1B (Figs 4, 5, and 6D). Overall, our data indicate that mitochondrial RNase H1 directly removes R-loops to restrict HR activity, which, in turn, helps maintain the proper mtDNA copy number during seed maturation and plant development.

The loss of RNase H1 led to embryonic lethality, defective mitochondria, and reduced mtDNA contents in mice due to the disruption of mtDNA replication [10,34,35]. We investigated whether this is also the case in *Arabidopsis* by examining DNA replication events using two-dimensional agarose gel electrophoresis (2D-AGE). Four fragments containing the repeats K1, I2, L2, or T2 were generated by enzyme digestion and used for the assay (Fig 8A and 8C). We detected replication intermediates in 3 regions, and in the K1 and T2 loci, the loss of mitochondrial RNase H1 had no effect on replication (Fig 8B and 8C). By contrast, in the region containing I2, replication increased in the absence of mitochondrial RNase H1. These results indicate that mitochondrial RNase H1 proteins might play different roles in mammals and *Arabidopsis*.

## OSB1 restricts HR by inhibiting R-loop formation at IRs

Several *Arabidopsis* mutants exhibit increased ectopic HR of IRs, including *msh1*, *osb1*, *recG1*, *recA2*, *recA3*, and *swib5* [20,24–28]. These proteins might function in different ways to regulate HR (reviewed in [17]). To further explore the pathways by which R-loops mediate HR, we looked for other potential regulatory factors in these mutants. Specifically, we assessed mitochondrial R-loops in *msh1*, *osb1*, and *recA3* (S7 Fig) by DRIP-qPCR. Our results showed that

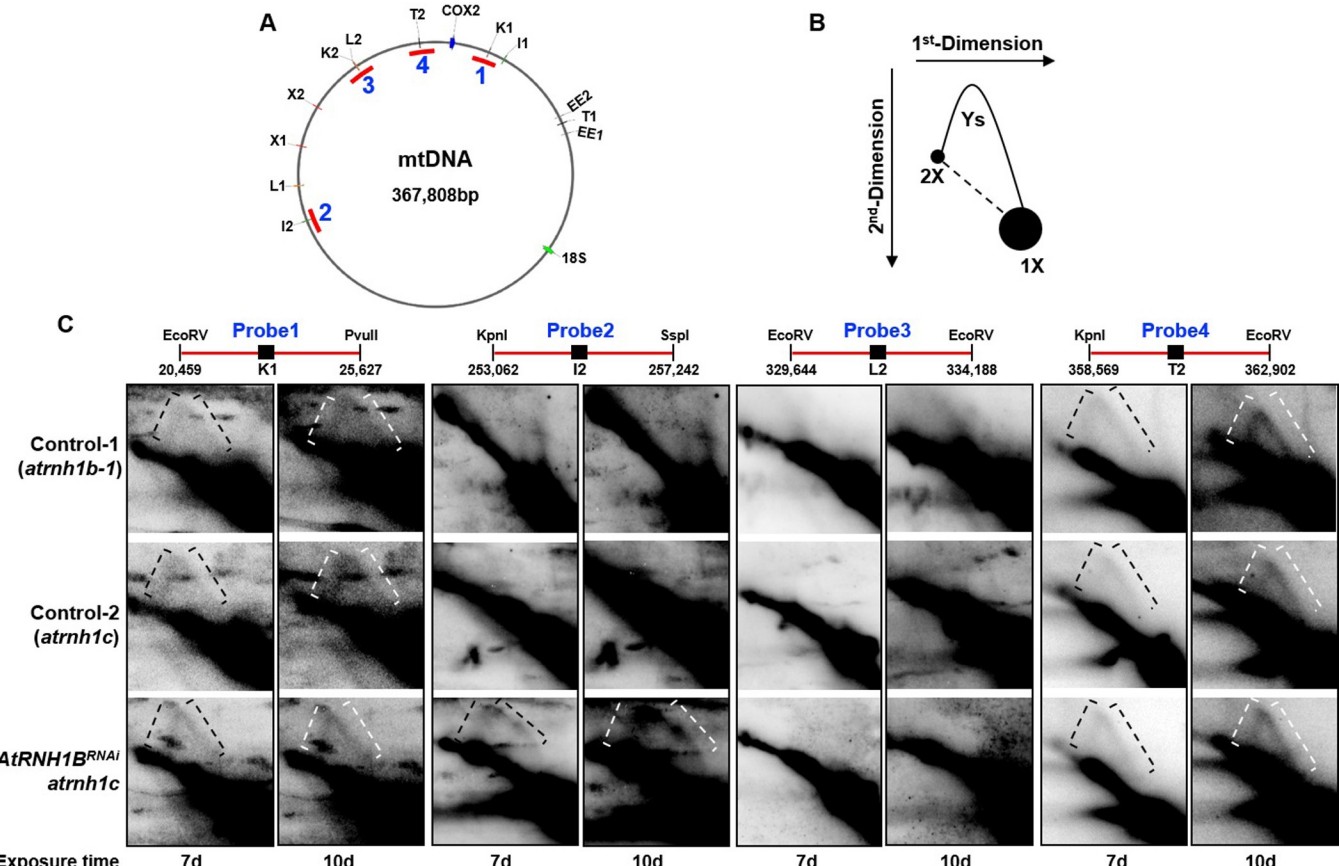

**Fig 8. Two-dimensional gel analysis detects replication events in the repeats.** (A) Schematic map of *Arabidopsis* mtDNA showing 6 repeat pairs. The fragments digested for 2D-AGE are indicated by red lines. (B) Illustration of the major 2D gel signals detected by the probe. (C) Two-dimensional gel analysis of digested mtDNA. Replication intermediates are indicated by square brackets. Probe (20,459–25,627) detects fragments containing repeat K1; Probe (253,814–256,094) detects fragments containing repeat I2; Probe (329,644–334,188) detects fragments containing repeat L2; and Probe (360,020–361,986) detects fragments containing repeat T2. The data underlying this figure can be found in S1 Raw Images. 2D-AGE, two-dimensional agarose gel electrophoresis; mtDNA, mitochondrial DNA; RNAi, RNA interference.

there were significantly more R-loops on repeats L and EE2 in *osb1* than in Col-0 and the other mutants, whereas repeat I did not accumulate R-loops (Fig 9A).

To decipher the relationship between R-loops and OSB1 during HR, we modulated the levels of mitochondrial R-loops in *osb1* by overexpressing *AtRNH1B* (Fig 9B and 9C) and quantified the HR frequency. Removing excessive R-loops by overexpressing *AtRNH1B* in *osb1* restored HR activity at repeats L and EE (L1/2 and E2/1) but had no significant effect on repeat I (Fig 9D). These results demonstrate that the role of OSB1 in regulating HR partially depends on the presence of R-loops at L and EE regions.

## Genetic interactions between *AtRNH1B* and OSB1

Finally, to confirm the genetic relationship between *AtRNH1B* and *OSB1*, we generated triple mutants by deleting *OSB1* in the *AtRNH1B*[RNAi] *atrnh1c* mutant background (S7D Fig). The *osb1 AtRNH1B*[RNAi] *atrnh1c* (hereafter named as *osb1 RNAi #1*) triple mutant showed more severe growth defects than *AtRNH1B*[RNAi] *atrnh1c* (Fig 10A), suggesting that a synergistic interaction occurs between AtRNH1B/1C and OSB1. We analyzed the R-loop levels in these

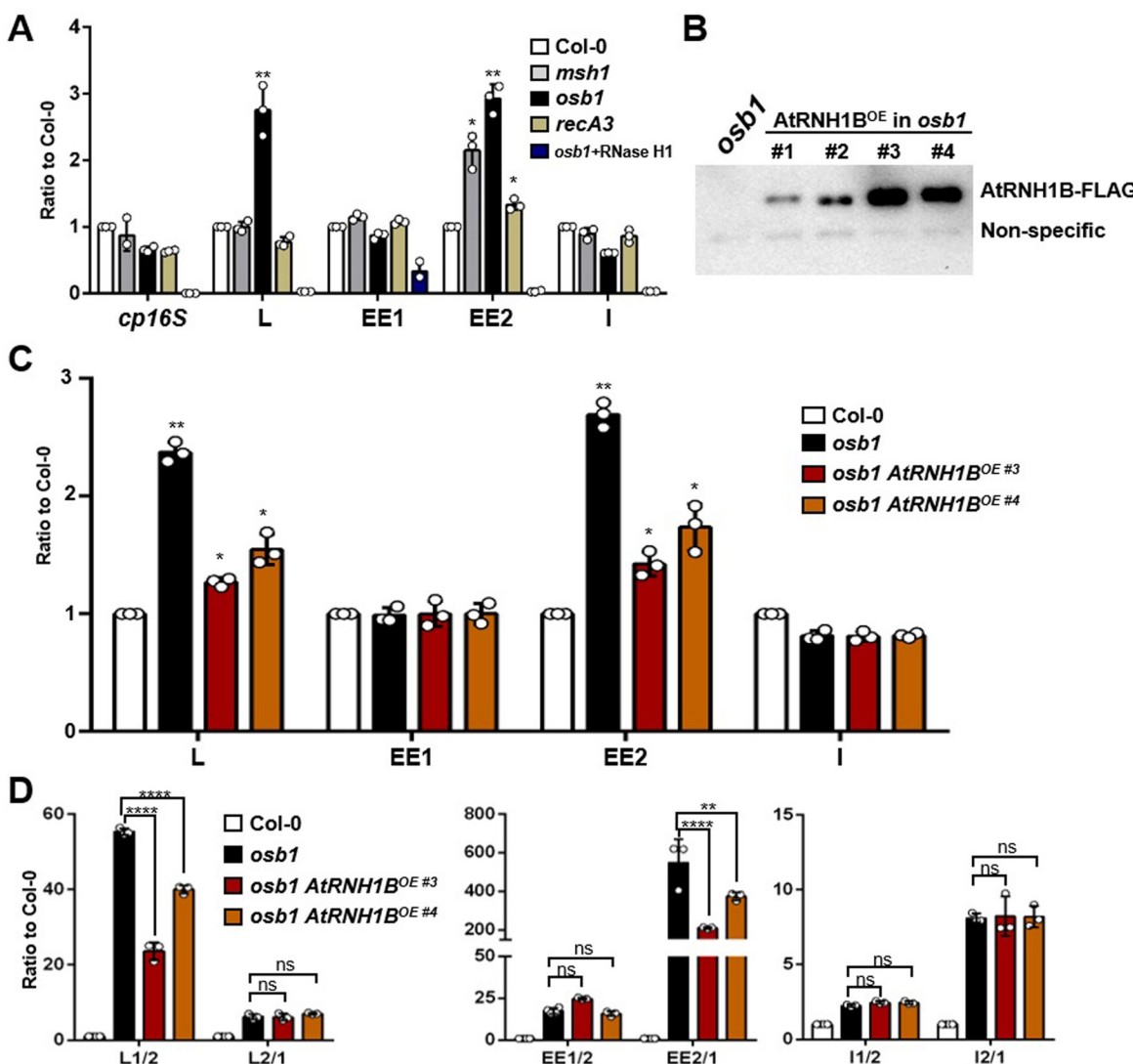

**Fig 9. OSB1 restricts HR by inhibiting R-loop formation at IRs. (A)** DRIP-qPCR of Col-0, *msh1*, *osb1*, and *recA3* seedlings. Data are normalized to Col-0 and shown as mean values ± SD; circles show the original data of 3 replicates. One-way ANOVA compared to Col-0 for each primer. $^*$, $p < 0.05$; $^{**}$, $p < 0.01$. **(B)** Immunoblot analysis of *35Spro:AtRNH1B-FLAG* (AtRNH1B$^{OE}$ #1- #4) in *osb1* using anti-FLAG antibody. *osb1* was used as a negative control. The unspecific band below AtRNH1B-FLAG is the loading control. **(C)** DRIP-qPCR of Col-0, *osb1*, and *osb1 AtRNH1B$^{OE}$* (#3 and #4) seedlings. One-way ANOVA compared to Col-0 for each primer. $^*$, $p < 0.05$; $^{**}$, $p < 0.01$. **(D)** qPCR of the crossover products (as depicted in A) of repeats L, EE, and I from Col-0, *osb1*, and *osb1 AtRNH1B$^{OE}$* (#3 and #4). One-way ANOVA compared to *osb1* for each primer. ns, no significance; $^{**}$, $p < 0.01$; $^{****}$, $p < 0.0001$. The data underlying this figure can be found in S1 Data and S1 Raw Images. ANOVA, analysis of variance; DRIP, DNA:RNA hybrid immunoprecipitation; HR, homologous recombination; IR, intermediate-sized repeat; OE, overexpression; OSB1, ORGANELLAR SINGLE-STRANDED DNA BINDING PROTEIN1; qPCR, quantitative PCR; SD, standard deviation.

mutants to determine their effects on R-loop formation. *AtRNH1B$^{RNAi}$ atrnh1c* showed an epistatic effect to *osb1* at repeat L and an additive effect at repeat EE (Fig 10B).

While detecting the HR frequency, we noticed that the fold change of crossover products in *AtRNH1B$^{RNAi}$ atrnh1c* was not as high as that in *atrnh1b/c* (Figs 7C and 10C), likely due to the residual AtRNH1B (S6B Fig). Consistent with the R-loop levels (Fig 10B), the levels of crossover products of L1/2 and L2/1 in the triple mutant were close to those in *AtRNH1B$^{RNAi}$ atrnh1c* at repeat L, indicating that OSB1 functions downstream of AtRNH1B. At repeats EE

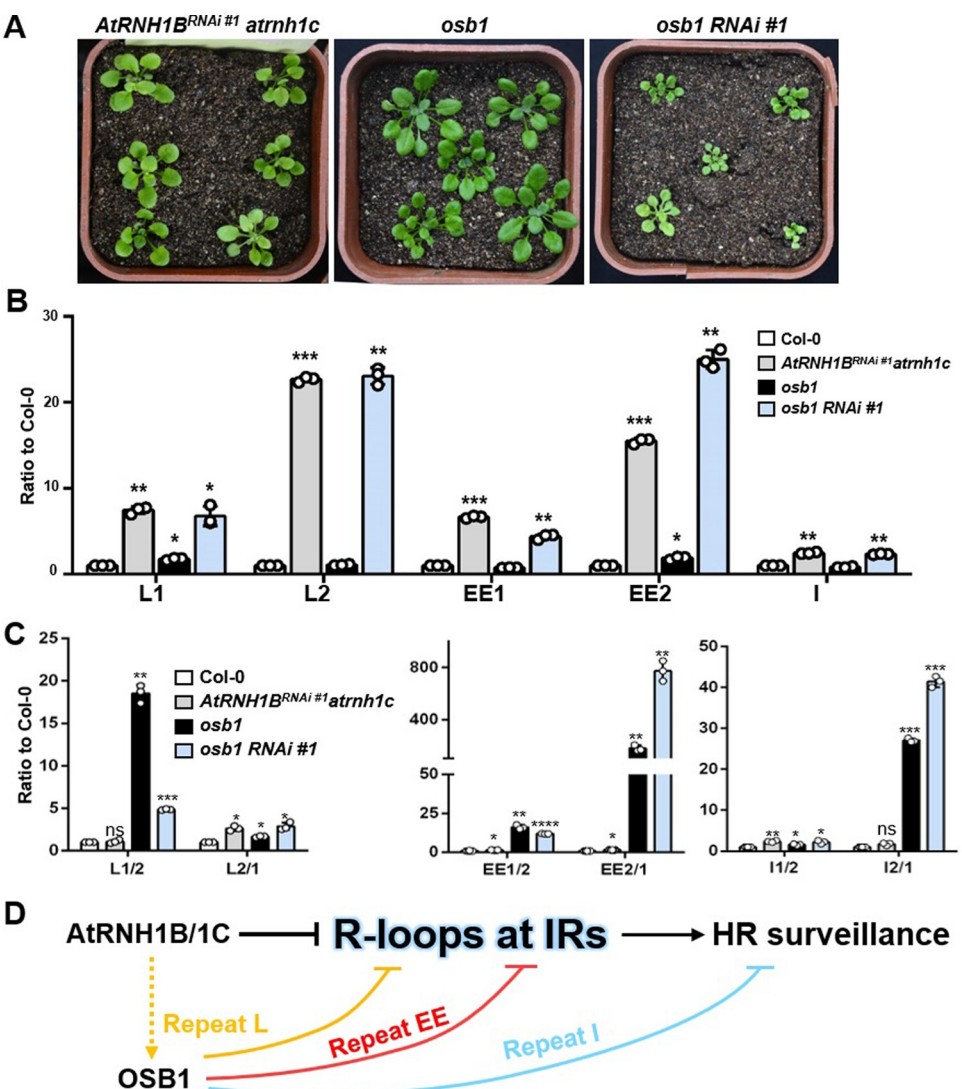

**Fig 10. The relationship between AtRNH1B/C and OSB1. (A)** Four-week-old *AtRNH1B^{RNAi #1} atrnh1c*, *osb1*, and *osb1 AtRNH1B^{RNAi #1} atrnh1c* plants (shown as *osb1RNAi #1*). Scale bars, 1 cm. **(B)** DRIP-qPCR of siliques from Col-0, *osb1*, *AtRNH1B^{RNAi #1} atrnh1c*, and *osb1 RNAi* #1. Data are normalized to Col-0 and shown as mean values ± SD; circles show the original data of 3 replicates. One-way ANOVA compared to Col-0 for each primer. $^{*}$, $p < 0.05$; $^{**}$, $p < 0.01$. **(C)** qPCR of the crossover products (as depicted in B) of repeats L, EE, and I from Col-0, *osb1*, *AtRNH1B^{RNAi #1} atrnh1c* and *osb1 RNAi* #1. One-way ANOVA compared to Col-0 for each primer. The data underlying this figure can be found in S1 Data. **(D)** R-loops trigged by AtRNH1B/1C and OSB1 mutation may promote HR of IRs. AtRNH1B/1C could repress mtDNA HR of IRs by modulating R-loops at IRs regions. Another HR surveillance factor, OSB1, controls HR via 3 different mechanisms at different IRs (indicated by 3 different colors). Specifically, OSB1 represses HR of repeat I independent R-loop and HR of repeat EE and L, which is partially dependent on inhibiting R-loop formation. In addition, AtRNH1B/1C has an epistatic effect to OSB1 at repeat L but a parallel effect to OSB1 at repeat EE. ANOVA, analysis of variance; DRIP, DNA:RNA hybrid immunoprecipitation; HR, homologous recombination; IR, intermediate-sized repeat; mtDNA, mitochondrial DNA; OSB1, ORGANELLAR SINGLE-STRANDED DNA BINDING PROTEIN1; qPCR, quantitative PCR; RNAi, RNA interference; SD, standard deviation.

and I, the triple mutant showed an additive HR frequency (EE2/1 and I2/1) compared to *osb1* (Fig 10C). In addition, the accumulation patterns of crossover products at repeats EE and I in *osb1* were different from those in *atrnh1b/c* (Figs 7C and 9D). Collectively, these results

indicate that OSB1 and AtRNH1B function in parallel for HR surveillance at repeats I and EE; at repeat L, AtRNH1B might function upstream of OSB1 for HR surveillance (Fig 10D).

## Discussion

The overaccumulation of R-loops in plants induces genome instability in both the nuclei and chloroplasts, and they must therefore be properly regulated [12,13,36,37]. Here, we found that the accumulation of R-loops in mitochondria also leads to genome instability by promoting ectopic mtDNA HR at IRs (Fig 10D). Unlike animals, which possess only 1 RNase H1 in charge of regulating R-loop levels in both the nucleus and mitochondrion, *Arabidopsis* contains 3 RNase H1 proteins that are thought to be localized to the nuclei, chloroplasts, and mitochondria, respectively. The chloroplast localized AtRNH1C is responsible for easing the head-on conflict between transcription and replication to maintain R-loop homeostasis, thus maintaining genome integrity in chloroplasts [12]. However, surprisingly, we determined that AtRNH1C was able to cover the RNase H1 function in the mitochondria of plants lacking AtRNH1B, providing a second layer of security to help maintain mitochondrial R-loop homeostasis. We also demonstrated that AtRN1HB functions collectively with the HR surveillance protein OSB1 to maintain mtDNA stability. These findings extend our understanding of R-loop functions and the mtDNA HR pathway.

The depletion of both AtRNH1B and AtRNH1C led to embryonic lethality (Figs 3 and S3 and S6), indicating that AtRNH1B is crucial for embryogenesis, which is in line with its consistent expression in pollen and seeds (Fig 2). AtRNH1B was also detected in other tissues and at different stages of development (Fig 2), suggesting that AtRNH1B plays important roles throughout the plant life cycle. Thus, the similar phenotypes of *AtRNH1B^RNAi^ atrnh1c* and *atrnh1c* during vegetative growth stage might have been due to the presence of residual AtRNH1B. Alternatively, perhaps AtRNH1B primarily functions in plant responses to development and/or genetic stresses that induce R-loop accumulation (S6 Fig).

In mammals, RNaseH1 is essential for removing RNA primers at the origin of replication for mtDNA, thus maintaining the stability of mtDNA [34,38,39]. The loss of RNase H1 in mice led to a phenomenon similar to that observed in *atrnh1b/c* [35]. However, we did not detect impaired mtDNA replication in these plants, and replication even increased in repeat I2 region (Fig 8C). These findings suggest that the function of RNase H1 in mitochondria is not completely conserved between mammals and plant. Indeed, 2 exonucleases with homology to the 5′ to 3′ exonuclease domain of *Escherichia coli* DNA polymerase I (5′ to 3′ EXO1 and 2) have been predicted to localize to organelle [19], suggesting that RNase H1 may not be required for the removal of RNA primers. However, it is still possible that RNase H1 affects mtDNA integrity by affecting replication in plants via other mechanisms, as the number of mitochondria in *atrnh1b/c* was reduced (S3F Fig). For example, RNase H1 might restrict aberrant R-loop formation to suppress origin-independent replication [40]. Alternatively, it might prevent transcription–replication conflict-triggered R-loop formation to promote replication [13]. These functions could explain the replication intermediates we detected in the I2 region (Fig 8C). Indeed, several studies have shown that the dysfunction of plant mitochondria and chloroplasts causes defects in embryogenesis [41–44]. DNA recombination is important for mtDNA maintenance [18], and large-scale rearrangements of organellar genomes could ultimately disturb the functions of mitochondria and/or plastids (reviewed in [30,45,46]). In *atrnh1b/c*, we indeed observed strongly enhanced HR at IRs and reduced mtDNA copy number (Fig 7 C–E).

Dual targeting is common among proteins involved in organellar gene regulation [47], such as RNA polymerase, DNA gyrase A, and DNA polymerases [48–50]. In general, the dual-targeted protein is delivered to the mitochondrion and chloroplast indiscriminately, although

the distribution between the protein subpopulations depends on the affinity of the receptors of each organelle for the sub-localization sequence [51]. Here, we demonstrated that the mitochondrial localization of AtRNH1C is normally completely suppressed (or to an undetectable level) in the presence of functional AtRNH1B, but is significantly activated when AtRNH1B is depleted (Fig 4), indicating that AtRNH1C is not a typical dual-targeted protein. While over 100 dual-targeted proteins have been identified [52], to our knowledge, facultative dual targeting that changes depending on cellular conditions is a newly discovered phenomenon in plants. Although we could not completely exclude the possibility that tiny (undetectable) AtRNH1C might exist in mitochondria, our data indicate that the dual localization is strictly regulated and could not be simply due to the increased expression of *AtRNH1C*, as AtRNH1C exclusively localized to chloroplasts when it was overexpressed in Col-0 (Fig 4A). Moreover, we determined that the signal sequence used for the mitochondrial targeting of AtRNH1C was the same as that used for chloroplast targeting and that this process was inhibited by the 69 to 77 aa fragment in this protein (Fig 5); further research is needed to elucidate the underlying mechanism.

In the absence of AtRNH1B, the levels of AtRNH1C increased as well (Fig 6). We speculate that when AtRNH1B levels are low in mitochondria, R-loops overaccumulate in the mtDNA, which, in turn, activate retrograde signaling to enhance the expression of *AtRNH1C* (Fig 6D). This mechanism could reflect the high degree of flexibility in the regulation of the organellar genome by the nucleus. The type of signaling that is activated and how the expression of *AtRNH1C* is regulated remain to be addressed. Considering the numerous proteins that could target both chloroplasts and mitochondria in plants [14,15], we suspect that this character will be found in other proteins.

In humans, R-loops promote the initiation of HR by recruiting factors involved in HR at double-strand breaks (DSBs), such as RAD52, RAD51, BRCA1, and BRCA2 [31]. Here, we demonstrated that R-loops can also promote HR, but whether the accumulation of downstream factors is altered is unknown. The underlying mechanism of mtDNA HR surveillance in *Arabidopsis* is not yet clear. Only a few genes have been shown genetically to repress ectopic HR of IRs, and their precise roles in HR repression remain unclear. The mutations of *E. coli* RecA-like proteins RECA2 and RECA3 cause increased HR of IRs, suggesting that a RecA-independent pathway might promote HR [26]. OSB1 is a plant-specific ssDNA binding protein that localizes to the mitochondria that is distinct from bacterium-type single-stranded DNA-binding proteins (SSB). The suppressor of HR could outcompete recombinase for the initial binding to ssDNA [25], thus simultaneously preventing R-loop formation, which is similar to the proposed function of mitochondrial ssDNA-binding protein (mtSSB) in mammals [11]. In order to maintain minimal heteroplasmy of mtDNA, mitochondria pass through a stringent genetic bottleneck in transmitting tissues [53,54]. In agreement with this process, *OSB1* is expressed in roots and gametophytic cells [25], which is highly similar to the expression pattern of *AtRNH1B* (Fig 2). The different ways in which OSB1 and AtRNH1B regulate HR of different IRs suggest that a rather complex mechanism regulates HR surveillance, but this mechanism requires further investigation.

In yeast and humans, the formation of R-loops around DSB sites promotes efficient HR-mediated DSB repair [31,32]. When organisms are exposed to DNA damaging agents, the HR pathway is important for DSB repair in organelles [17,55]. The response to genotoxic stress that enhances HR between IRs is controlled by proteins involved in repair, including RECA HOMOLOG3 (RECA3), WHIRLY2 (WHY2), ODB1, and DNA POLYMERASE 1B (POL1B) [26,56,57]. Multiple factors promote recombination processes during mtDNA DSB repair, but the complete set of factors involved in these processes remains to be identified. Our results

indicate that R-loop levels are important for maintaining proper HR activity, which might also be involved in HR-mediated DNA repair.

In conclusion, our findings indicate that mitochondrial genome stability is safeguarded by RNase H1 proteins. AtRNH1C can function as a mitochondrial RNase H1 protein when cells lack AtRNH1B. The mutation of both AtRNH1B and AtRNH1C results in R-loop accumulation and increased ectopic HR frequency, leading to decreased mtDNA copy number and ultimately resulting in aborted embryogenesis in plants. Our findings provide new insight into mitochondrial genome maintenance during development and in response to environmental changes, as well as the tight communication among organelles.

## Methods

### Plant materials and growth conditions

Seeds of wild-type *A. thaliana* ecotype Col-0 (Columbia-0) and T-DNA insertion mutants *atrnh1b-1* (SALK_141349), *atrnh1b-2* (SALK_020028), *atrnh1c* (SAIL_97_E11), and *msh1* (SALK-046763) were obtained from the Nottingham Arabidopsis Stock Centre, United Kingdom. The primers used for genotyping are listed in S1 Table. The plant CRISPR/Cas-9 system [12,58] was used to generate genomic deletions of *OSB1* (*osb1*) and *RECA3* (*recA3*). The sequences of the 2 sgRNAs are listed in S1 Table, and the locations of these sgRNAs are indicated in S8 Fig. Surface-sterilized seeds were placed on MS medium (Murashige and Skoog salt base; Sigma-Aldrich, St. Louis, MO, USA) containing Phytagel (0.25%, w/v) supplemented with 1% (w/v) sucrose and incubated at 4˚C for 2 days for stratification. The plants were grown under long-day conditions at 22˚C under a 16-hour light/8-hour dark cycle.

### Transgenic plants production

For the complementation experiments, *AtRNH1B* and *AtRNH1C* genomic DNA sequences from 2-kb upstream of ATG to 1-kb downstream of TAG were cloned into pCambia1300, and the GUS/GFP/FLAG tags were fused to the N-termini. The vectors were constructed using the Fast-Cloning method [59], and the primers are listed in S1 Table.

For RNAi experiment, the 3′ UTR of AtRNH1B cDNA was amplified with the following primers: 1B_RNAi(*Xho*I)-F CCGctcgagGGCTTAAAGGATATTATCGGTG, 1B_RNAi(*Bgl*‖)-R GGAagatctTTTCGTCTTAAATCCTTTATAAGA. Then the fragments were inserted to pUCCRNAi vector with *Xho*I and *Bgl*II and *Sal*I and *Bam*HI (isoschizomers of *Xho*I and *Bgl*II) through enzyme digestion and T4 ligation, respectively. Next, the pUCCRNAi_AtRNH1B vector was digested with *Pst*I and the fragment containing St-GA20oxiIN flanked with forward and reverse AtRNH1B fragments was inserted to pCambia2300 vector with 35S promoter.

These binary vectors were transformed into Agrobacterium tumefaciens strain GV3101. *Arabidopsis* transformation was performed by the floral dip method [60], and independent single insertion lines were identified for each transgene.

### Expression and purification of recombinant proteins

The coding sequence of *AtRNH1B* (or the sequence with the site mutation D191N) without the mitochondrial targeting sequence was cloned into pGEX4T-1 and expressed in *E. coli* BL21 (DE3) cells. The cells were grown at 37˚C until the OD600 reached 0.6 to 0.8. After adding IPTG to a final concentration of 0.1 mM, the cells were transferred to 16˚C and incubated overnight with gentle shaking. The cells were collected, resuspended in 1× PBS buffer, and immediately sonicated until the suspension became transparent. The supernatant was collected, and the protein purified using GST beads (88822, Thermo Fisher Scientific, Rockford,

IL, USA) at 4°C for 2 hours. The beads were washed 3 times with 1x PBS buffer and the proteins eluted with elution buffer (50 mM reduced glutathione in 1x PBS buffer).

## RNase H1 activity analysis

To analyze RNase H1 activity, an enzymatic reaction was performed in Reaction Buffer (50 mM KCl, 20 mM HEPES-KOH pH 7.0, 4% glycerol, 1 mM DTT, 50 μg/ml bovine serum albumin [BSA], 4 mM $MgCl_2$) with 100 nM FAM (5-carboxyfluorescein)-labeled RNA:DNA substrate and 2 nM to 2 μM purified AtRNH1B/1BM in a reaction volume of 10 μl. The reaction was conducted for 5 to 30 minutes at 37°C and stopped by the addition of EDTA to a final concentration of 40 mM. The products were analyzed on 12% native TBE-polyacrylamide gels and visualized by fluorescence readout.

## Mitochondrion and chloroplast fractionation

Mitochondrion fractionation was performed as previously described [61]. About 150 3-week-old seedlings were homogenized in 120-ml Grinding Medium (0.3 M sucrose, 25 mM tetrasodium pyrophosphate, 1% (w/v) polyvinylpyrrolidone-40, 2 mM EDTA, 10 mM $KH_2PO_4$, 1% (w/v) BSA, 20 mM sodium L-ascorbate, 1 mM DTT, 5 mM cysteine, pH 7.5). The homogenate was filtered through 4 layers of Miracloth and centrifuged at 2,500 g for 5 minutes, and the resulting supernatant was then centrifuged at 20,000 g for 15 minutes. The pellet was resuspended in 40-ml Washing Medium (0.3 M sucrose, 10 mM TES, pH 7.5), and repeated 1,500 g and 20,000 centrifugation steps. The resulting pellet was gently resuspended in 1- to 2-ml Washing Medium and fractionated on a Percoll step gradient (18% to 27% to 50%) by centrifugation for 45 minutes at 40,000 g. Mitochondria were collected at the 27% to 50% interface and diluted with Washing Medium. After centrifugation at 35,000 g for 15 minutes, the mitochondrial pellet was collected and stored.

Chloroplast fractionation was performed as previously described [12]. About 80 3-week-old seedlings was homogenized in chloroplast isolation buffer (CIB) (10 mM HEPES-KOH (pH 8.0), 150 mM sorbitol, 2.5 mM EDTA (pH 8.0), 2.5 mM EGTA (pH 8.0), 2.5 mM $MgCl_2$, 5 mM $NaHCO_3$, and 0.1% BSA). The homogenate was filtered through 3 layers of Miracloth and centrifuged at 200 g for 3 minutes. The resulting supernatant was then centrifuged at 1,700 g for 10 minutes. The pellet was resuspended in 1 ml CIB and fractionated on a Percoll step gradient (40% to 80%) by centrifugation for 30 minutes at 1,500 g. The intact chloroplasts were taken from the interface between the 40% and 80% Percoll and washed twice in CIB buffer without BSA.

## Protein extraction and immunoblot analysis

The powdered leaf tissue, chloroplasts, and mitochondria were lysed in buffer containing 50 mM Tris-HCl, pH 7.5, 150 mM NaCl, 5 mM EDTA, 1 mM DTT, 1 mM PMSF, 0.2% Triton X-100, 10% (v/v) glycerol, 1x protease inhibitor (Sigma-Aldrich). Protein extracts were denatured, separated on an SDS-PAGE, and transferred onto a polyvinylidene difluoride (PVDF) membrane. The membrane was blocked in 5% nonfat milk in 1× TBST, probed with the indicated primary antibodies, and then incubated with anti-rabbit or anti-mouse secondary antibodies conjugated with horseradish peroxidase. The signals were detected with a chemiluminescence reaction using SuperSignal West Dura Extended Duration Chemiluminescent Substrate (Thermo Fisher Scientific). The purity of the chloroplast and mitochondrial fractions was monitored using antibodies against nuclear H3, chloroplast psbO, and mitochondrial IDH1. The antibodies used were the following: Anti-FLAG (1:5,000 dilution; Sigma-Aldrich), Anti-psbO (1:5,000 dilution; Abcam, Cambridge, UK),

Anti-IDH1 (1:2,500; PhytoAB, San Jose, CA, USA), Anti-H3 (1:5,000 dilution; ABclonal, Wuhan, Hubei, China), Anti-GUS (1:2,500 dilution; Abcam), Anti-Mouse (1:10,000 dilution; EASYBIO, Beijing, China), and Anti-Rabbit (1:10,000 dilution; EASYBIO).

### Seed clearing and microscopy

Seed clearing was performed as previously described [62]. In brief, developing siliques at various stages were fixed in ethanol:acetic acid (9:1) and washed with 70% ethanol. The seeds were isolated, mounted on slides in clearing solution (glycerol/chloral hydrate/water in a ratio of 1:8:3), and observed under an Olympus BX51 microscope (Olympus, Tokyo, Japan).

### GUS staining

GUS staining was performed as described by [63]. Developing siliques were cut longitudinally and fixed in ice-cold 90% acetone for 1 hour at −20˚C. After washing 3 times with PBS, the tissue was immersed in staining solution containing including X-gluc and vacuum-infiltrated for 20 minutes. After staining for 12 hours at 37˚C, the samples were mounted on slides in clearing solution and observed under an Olympus BX51 microscope.

### RNA extraction and reverse transcription

Total RNA was extracted from the leaves of 14-day-old plants using TRIzol (Ambion, Carlsbad, CA, USA) and treated with DNase I for 15 minutes. cDNA was synthesized from the RNA using a PrimeScript RT reagent Kit with gDNA Eraser (Takara, RR047A, Shiga, Japan).

### Protoplast transformation

The plasmids used for protoplast transformation were constructed by cloning the coding sequences of the genes into pUC19 vectors containing the 35S promoter and eGFP tag.

Protoplast transformation was performed as previously described [64]. Briefly, protoplasts were extracted from the leaves of 4-week-old plants and transformed via 20% PEG-Ca-mediated transformation. After 14 hours of transient expression, GFP fluorescence and chlorophyll auto-fluorescence were detected under a confocal microscope (Zeiss LSM780, Zeiss, Oberkochen, Germany). The mitochondria were stained with MitoTracker Red CMXRos (Thermo Fisher Scientific, Cat. No. M7512).

### Immunogold labeling and electron microscopy

Samples for transmission electron microscopy were prepared using traditional chemical fixation [65]. A total of 3 to 4 leaves were fixed in 2% (v/v) glutaraldehyde and 0.5% (w/v) paraformaldehyde in 50 mM PBS for 2 hours at room temperature (RT), rinsed 3 times in 50 mM PBS, dehydrated, infiltrated, and embedded in LR White resin (Sigma-Aldrich). Resin blocks were polymerized at 48˚C for 48 hours and serially sectioned ultrathin (approximately 80-nm thick). Ultrathin sections were blocked and subsequently incubated in a moist chamber with primary antibody diluted 1:150 (anti-GFP; Abcam Ab290) 2 hours at RT. Secondary antibody that conjugated to 10-nm gold particles (Sigma-Aldrich; G7402) were diluted 1:50 with PBS/BSA and incubated for 1 hour at RT. Sections were extensively washed, stained with uranyl acetate, and examined with FEI Tecnai Spirit D1297 transmission electron microscope (FEI, Hillsboro, OR, USA).

## Chromatin immunoprecipitation analysis of mitochondria

Crude mitochondria were extracted from AtRNH1B/1C-GFP complementation plants (Col-0 as a control) as previously described [66]. Briefly, 5 g of 14-day-old seedling tissue was homogenized in Grinding Medium (with 0.8% BSA and 0.1% β-mercaptoethanol) on ice. After filtering through Miracloth (Millipore, Billerica, MA, USA), the homogenate was centrifuged at 1,800 g for 7 minutes. The supernatant was transferred to a new tube and centrifuged at 20,000 g for 20 minutes at 4˚C. The pellet was collected as crude mitochondria for ChIP following a published protocol [63]. The crude mitochondria were suspended with Grinding Medium and cross-linked in 1% formaldehyde. After stopping the cross-linking by adding 0.125 M glycine, the sample was centrifuged and the pellet was collected. The mitochondrial pellet was suspended in lysis buffer (50 mM Tris-HCl, pH 8.0, 10 mM EDTA, 1% SDS, and 1x protease inhibitor) and sonicated 3 times for 10 minutes each time (30-second on/off intervals) using a Bioruptor Pico (Diagenode, Liege, Belgium). The sample was centrifuged at maximum speed for 10 minutes. The supernatant was collected and diluted 10-fold with ChIP dilution buffer (1.1% Triton X-100, 1.2 mM EDTA, 16.7 mM Tris-HCl, pH 8.0, 167 mM NaCl). A 1-ml aliquot of each sample was incubated with 2-μl anti-GFP antibody (Abcam, ab290) overnight at 4˚C with rotation. Moreover, 50-μl Dynabeads Protein G magnetic beads (Invitrogen, 10004D, Carlsbad, CA, USA) that were washed with ChIP dilution buffer were added to the sample. After 3 hours of incubation with rotation, the beads were washed sequentially with Low Salt Buffer, High Salt Buffer, LiCl Buffer, and TE Buffer (for each buffer, the beads were rinsed quickly, followed by a 5-minute wash). The immune complexes were eluted by adding 100 μl 10% Chelex resin (Bio-Rad, 1422842, Hercules, CA, USA) and incubating at 95˚C for 10 minutes at 1,300 rpm. The sample was cooled to RT and incubated with 2-μl Proteinase K (10 mg/ml, Amresco 0706) for 1 hour at 45˚C. After boiling the sample for 10 minutes at 95˚C and centrifuging at 1,300 rpm, the eluted sample was purified with phenol chloroform prior to sequencing.

## DNA:RNA hybrid immunoprecipitation

DRIP analysis was performed as described previously [67]. Briefly, total DNA was extracted from the samples as described above without extracting the nuclei, and the genomic DNA was digested into approximately 100-bp fragments using MseI, DdeI, AluI, and HpaII (New England Biolabs, Ipswich, MA, USA). Extracted DNA with or without RNase H (New England Biolabs) treatment was used in the DRIP assay. The DNA was incubated with purified S9.6 antibody overnight, followed by incubation with Protein G beads, washing, and DNA recovery.

## Two-dimensional agarose gel electrophoresis

Two-dimensional AGE was carried out as described previously [68]. mtDNA was purified from mitochondria extracted as described above. A total of 100-μg mitochondrial nucleic acid sample was digested with 40 U of the indicated restriction enzymes in a 200-μl reaction volume for 4 hours at 37˚C. The DNA was precipitated with isopropanol and loaded onto a 0.4% agarose without ethidium bromide. First-dimension gels were run at 0.7 V/cm for 26 hours at RT. The DNA-containing lanes were excised and rotated 90˚ counterclockwise, and 1% agarose containing 0.3 μg/ml ethidium bromide was cast around the first-dimension gel slices. Second-dimension gels were run at a constant 260 mA for 6 hours at 4˚C and subjected to DNA gel blotting. The blots from the 2D gels were hybridized to radiolabeled probes produced with a Random Primer DNA Labeling Kit Ver. 2 (Takara, 6045) for specific regions of mtDNA, as indicated above. The autoradiographs were exposed to phosphor screens for 7 to 10 days, and the phosphor screens were scanned using a phosphorimager.

## qPCR

qPCR was performed using a LightCycler 480 (Roche, Basel, Switzerland). The reaction mixture (11 μl) contained 0.5-μl DNA, 5.5 μl of 480 SYBR Green I Master (Roche), and 2.5 pmol each of forward and reverse primer of interest (see S1 Table for primer sequences). To quantify mtDNA copy number, the qPCR results were normalized against the nuclear genes *ACT2* and *UBC*. The accumulation of mtDNA recombination products was assessed using primers flanking each repeat, as depicted in Fig 5B. The mitochondrial single-copy loci *COX2* (AtMG00160) and *18S rRNA* (AtMG01390) were used for normalization.

## Statistical analysis

Data were processed using Prism v.6 (GraphPad Software). Statistical analysis was performed using 1-way ANOVA. For qPCR, 1-way ANOVA was performed to compare to the expression level of each gene to that in Col-0. ns, no significance; *, $p < 0.05$; **, $p < 0.01$; ***, $p < 0.001$; ****, $p < 0.0001$. The uncertainty in the mean is reported as the standard deviation (SD) of the mean.

## Supporting information

**S1 Fig. RNase H activity of AtRNH1B in vitro. (A)** Structure of purified GST-fused proteins. GST tag was fused to the N terminus of AtRNH1B without the MTS (GST-AtRNH1B). The Asp (191 D) was mutated to Asn (N) in GST-AtRNH1B fusion protein (GST-AtRNH1BM). **(B)** SDS-PAGE of purified recombinant proteins. The molecular weight of GST and GST-AtRNH1B/M are approximately 25 kDa and approximately 63 kDa, respectively. **(C)** Related to Fig 1D, the amounts of proteins used in Coomassie blue staining are 10 times of purified protein used in RNase H activity assay. **(D)** The RNA:DNA hybrid substrate with FAM-labeled RNA (100 nM) following 5 minutes of incubation with increasing concentrations of GST-AtRNH1B and GST-AtRNH1BM. **(E)** Microscopic observation of GUS signals in epidermal and mesophyll cells of *AtRNH1Bpro:AtRNH1B-GUS atrnh1b-1* after GUS staining. For comparison, the cells were selected at same area through changing focal length. Col-0 was included as negative control to indicate the cell without GUS activity, which was shown orange-red without blue GUS signals as the contrast under microscopy. Scale bars, 100 μm. The data underlying this figure can be found in S1 Raw Images. GST, glutathione-S-transferase; GUS, β-glucuronidase enzyme; SDS-PAGE, sodium dodecyl sulfate-PAGE. (PPTX)

**S2 Fig. *atrnh1b* mutants have no obvious phenotype. (A)** Gene structure of *AtRNH1B* genomic DNA. White boxes represent UTRs, black boxes represent exons, and black lines represent introns. The triangles point to the positions of T-DNA insertions in *atrnh1b-1* and *atrnh1b-2*. **(B, C)** Twenty-eight cycles of RT-PCR of *AtRNH1B* transcript in *atrnh1b-1* (B) and *atrnh1b-2* (C); primers are shown in A. *GAPDH* was used as the reference gene. **(D)** Col-0 and *atrnh1b* plants at the vegetative stage (upper) and reproductive stage (bottom). Scale bars, 1 cm. The data underlying this figure can be found in S1 Raw Images. RT-PCR, reverse transcription PCR; UTR, untranslated region. (PPTX)

**S3 Fig. The *atrnh1b/c* mutant is embryo lethal. (A)** Representative siliques of *atrnh1b-1* and *atrnh1c* plants, and siliques from reciprocal crosses of Col-0 and *atrnh1b 1c$^{+/-}$*. Scale bars, 200 μm. **(B)** The ratio of *atrnh1b 1c$^{+/-}$* to *atrnh1b-1* was examined in 50 plates, each containing approximately 150 plants. Seeds were harvested from *atrnh1b 1c$^{+/-}$* plants, sown on 1/2 MS medium, and selected based on Basta resistance. Because the *atrnh1c* mutant is resistant to

Basta, the live plants are *atrnh1b 1c⁺/⁻*, and the dead plants are *atrnh1b-1*. **(C)** F2 plants of *atrnh1b 1c⁺/⁻* complemented with *AtRNH1Bpro:AtRNH1B/AtRNH1BΔMTS-GFP* that were sown on 1/2 MS medium and selected based on Basta and hygromycin resistance. The complemented plants are resistant to hygromycin. The green plants are heterozygous *atrnh1b 1c⁺/⁻*, and the yellow plants are homozygous *atrnh1b/c*. **(D)** Different phenotypes of embryos in abnormal seeds that failed to transition to the heart stage. Scale bars, 20 μm. **(E)** Transmission electron microscopy of globular embryos from Col-0 and *atrnh1c* plants. White arrowheads indicate internal cristae membranes. Scale bars, 500 nm. **(F)** Confocal microscopy of mitochondria (labeled with MitoTracker) in *atrnh1b/c* aborted seeds and Col-0 normal seeds. Magenta = MitoTracker. Scale bars, 10 μm. The data underlying this figure can be found in S1 Data. 1/2 MS, half strength Murashige & Skoog (MS); GFP, green fluorescent protein; MTS, mitochondrial targeting signal.
(PPTX)

**S4 Fig. AtRNH1C can function as mitochondrial RNase H1 in cells lacking AtRNH1B. (A)** Protoplasts from the root tips of *AtRNH1C-GFP* transgenic plants in *atrnh1b-1* background. **(B)** Sequence alignment of AtRNH1B and AtRNH1C. Red boxes show the 2 deletions in Figs 5 and S5C. Domains are shown above the sequences. **(C)** Transformed protoplasts from Col-0. **(D)** GUS staining of different transgenic lines (#1, #2, #3, and #6) of *AtRNH1Cpro:AtRNH1C-GUS* in the *atrnh1b-1* and *atrnh1c* backgrounds, respectively. CTS, chloroplast targeting signal; GFP, green fluorescent protein; GUS, β-glucuronidase enzyme; HBD, hybrid binding domain; MTS, mitochondrial targeting signal.
(PPTX)

**S5 Fig. Knocking down *AtRNH1B* results in sterility in *atrnh1c*. (A)** Carboxyl terminus of *AtRNH1B* cDNA. Blue box indicates the RNAi target sequence. **(B)** RT-qPCR analysis indicates the reduced expression of *AtRNH1B* in different *AtRNH1Bᴿᴺᴬⁱ atrnh1c* lines (hereafter, shown as #1–8 for briefness); primers are indicated in A. Data are normalized to *atrnh1c* and shown as mean values ± SD; circles show the original data of 6 repeats from 2 biological replicates. **(C)** Seedlings of 3-week-old Col-0, *atrnh1b-1*, *atrnh1c*, and *AtRNH1Bᴿᴺᴬⁱ atrnh1c* #1, #2, and #3. **(D)** Branches of Col-0 and *AtRNH1Bᴿᴺᴬⁱ atrnh1c* plants. **(E)** Siliques of Col-0 (lower) and *AtRNH1Bᴿᴺᴬⁱ atrnh1c* #1 (upper). Scale bars, 1 cm. **(F)** Transmission electron microscopy of siliques from Col-0, *atrnh1c*, and *AtRNH1Bᴿᴺᴬⁱ atrnh1c* plants. Scale bars, 1 μm. **(G)** Pollen viability in 5-week-old plants by Alexander staining. Scale bars, 10 μm. **(H)** Siliques from reciprocal crosses of Col-0 and *AtRNH1Bᴿᴺᴬⁱ atrnh1c* #1. Scale bars, 200 μm. The data underlying this figure can be found in S1 Data. RNAi, RNA interference; RT-qPCR, reverse transcription quantitative PCR; SD, standard deviation.
(PPTX)

**S6 Fig. Changes in mtDNA copy number and HR frequency during development. (A)** Schematic map of *Arabidopsis* mtDNA. Six repeat pairs are indicated in the map. **(B)** Relative quantification of copy numbers of mtDNA (*18S*, *COX2*) gene sequences in different tissues of Col-0. Nuclear genes *UBC* and *ACT2* were used as reference genes. **(C)** Relative quantification of the crossover products (as depicted in Fig 5B) of L and EE in different tissues of Col-0. Mitochondrial genes *18S* and *COX2* were used as reference genes. Data are normalized to rosette leaves and shown as mean values ± SD; circles show the original data of 6 repeats containing 2 biological replicates. One-way ANOVA compared to rosette leaves for each locus. *, $p < 0.05$; **, $p < 0.01$; ***, $p < 0.001$; ****, $p < 0.0001$. others not shown are not significant. The data underlying this figure can be found in S1 Data. *ACT2, ACTIN 2*; ANOVA, analysis of variance; *COX2, CYTOCHROME OXIDASE 2*; HR, homologous recombination; mtDNA, mitochondrial

DNA; SD, standard deviation; *UBC, UBIQUITIN-CONJUGATING ENZYME*.
(PPTX)

**S7 Fig. *osb1* and *recA3* mutants. (A, B)** Gene structures of *OSB1* (A) and *RECA3* (B) genomic DNA. gRNAs for CRISPR knockout are indicated as red lines. **(C)** DNA gel shows the results of CRISPR knockout. Primers for PCR are shown as *osb1*-F/R and *recA3*-F/R in (A). The size of PCR products with *osb1*-F+R is 342 bp in Col-0 and 163 bp in *osb1* and *osb1 AtRNH1B^{RNAi} atrnh1c* (*shown as osb1 RNAi* #1 for briefness). The size of PCR products with *recA3*-F+R is 532 bp in Col-0 and 240 bp in *recA3*. **(D)** Sequence chromatograms showing the CRISPR/Cas-9–mediated deletion of *OSB1*. **(E)** Sequence chromatograms showing the CRISPR/Cas-9–mediated deletion of *RECA3*. The data underlying this figure can be found in S1 Raw Images. Cas-9, CRISPR associated protein-9; CRISPR, clustered regularly interspaced short palindromic repeats; F, forward; gRNA, guide RNA; *OSB1*, ORGANELLAR SINGLE-STRANDED DNA BINDING PROTEIN1; R, reverse.
(PPTX)

**S1 Movie. *Arabidopsis* AtRNH1B is a mitochondrial RNase H1 protein.** LiTone LBS Light-sheet microscopy of root of *AtRNH1Bpro*:*AtRNH1B-GFP atrnh1b* transgenic plants. Scale bars, 10 μm. Green = GFP and magenta = MitoTracker.
(PPTX)

**S2 Movie. *Arabidopsis* AtRNH1B is expressed in both epidermal and mesophyll cells.** OLYMPUS DP80 microscope visualizes the GUS signals from the top surface to the bottom of the leaf carrying the *AtRNH1Bpro*:*AtRNH1B-GUS* in *atrnh1b* mutant (complementation lines).
(PPTX)

**S1 Table. Primers used in this study.**
(XLSX)

**S1 Data. Values for all data used to create the graphs throughout the paper.**
(XLSX)

**S1 Raw Images. Unprocessed images of all gels and blots in the paper.**
(PPTX)

# Acknowledgments

We thank the Core Facility of the Center of Biomedical Analysis in Tsinghua University for help with confocal microscopy analysis. The Sun Lab is supported by Tsinghua-Peking Center for Life Sciences.

# Author Contributions

**Conceptualization:** Qianwen Sun.

**Data curation:** Lingling Cheng, Wenjie Wang, Yao Yao.

**Formal analysis:** Lingling Cheng, Wenjie Wang, Yao Yao, Qianwen Sun.

**Investigation:** Lingling Cheng, Wenjie Wang, Yao Yao.

**Methodology:** Lingling Cheng, Wenjie Wang, Yao Yao.

**Project administration:** Qianwen Sun.

**Supervision:** Qianwen Sun.

**Writing – original draft:** Lingling Cheng, Wenjie Wang, Qianwen Sun.

**Writing – review & editing:** Wenjie Wang, Qianwen Sun.

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
