## [Editor Report · Decision Letter 0]

15 Jan 2021

Dear Dr Sun, 

Thank you for submitting your manuscript entitled "A facultative dual-targeting mechanism maintains mitochondrial RNase H1 and R-loop homeostasis for genome integrity and early embryogenesis in Arabidopsis" for consideration as a Research Article by PLOS Biology. Please accept my apologises for the delay in getting back to you as we were consulting with an academic editor about your submission.

Your manuscript has now been evaluated by the PLOS Biology editorial staff as well as by an academic editor with relevant expertise and I am writing to let you know that we would like to send your submission out for external peer review.

Please re-submit your manuscript within two working days, i.e. by Jan 17 2021 11:59PM.

Kind regards,

Richard Hodge, PhD 

Associate Editor

PLOS Biology

---

## [Decision Letter · Decision Letter 1]

16 Feb 2021

Dear Dr Sun,

Thank you very much for submitting your manuscript "A facultative dual-targeting mechanism maintains mitochondrial RNase H1 and R-loop homeostasis for genome integrity and early embryogenesis in Arabidopsis" for consideration as a Research Article at PLOS Biology. Your manuscript has been evaluated by the PLOS Biology editors, an Academic Editor with relevant expertise, and by three independent reviewers.

The reviews are attached below. You will see that the reviewers find your conclusions novel and interesting, but they also raise concerns with the presentation and interpretation of the data in the manuscript. Reviewer 3, in particular, thinks that the findings do not clarify the role of RNase H1 or R-loops in mitochondrial function, and raises several points that need to be clarified, some of them requiring more analyses.

In light of the reviews (attached below), we will not be able to accept the current version of the manuscript, but we would welcome re-submission of a revised version that takes into account the reviewers' comments. We cannot make any decision about publication until we have seen the revised manuscript and your response to the reviewers' comments. Your revised manuscript is also likely to be sent for further evaluation by the reviewers.

We expect to receive your revised manuscript within 3 months.

**IMPORTANT - SUBMITTING YOUR REVISION**

3. Resubmission Checklist

a) *Published Peer Review*

b) *PLOS Data Policy*

d) *Blurb*

Please also provide a blurb which (if accepted) will be included in our weekly and monthly Electronic Table of Contents, sent out to readers of PLOS Biology, and may be used to promote your article in social media. The blurb should be about 30-40 words long and is subject to editorial changes. It should, without exaggeration, entice people to read your manuscript. It should not be redundant with the title and should not contain acronyms or abbreviations. For examples, view our author guidelines: https://journals.plos.org/plosbiology/s/revising-your-manuscript#loc-blurb

Sincerely,

Ines

--

Ines Alvarez-Garcia, PhD

Senior Editor,

PLOS Biology

on behalf of

Richard Hodge,

Associate Editor,

rhodge@plos.org,

PLOS Biology

Reviewers’ comments

Rev. 1:

This is a well-written and thoroughly presented manuscript reporting on the role of RnaseH in maintaining organelle genome integrity in Arabidopsis, and it's expression levels and role in embryogenesis. The experiments and resulting data are clear and the presentation and discussion of the results are founded in the data. I found the work to be convincing, and this provides a very helpful addition to our knowledge of mechanisms involved in replication and maintenance organelle genomes during plant development. I have two minor points as follows.

1. In the introduction the size of plant mitochondrial genomes is mentioned. In recent years much larger mitochondrial genomes have been characterized in some angiosperms, particularly Silene. PNAS August 18, 2015 112 (33) 10185-10191; first published May 5, 2015; https://doi.org/10.1073/pnas.1421397112 This could be updated in your manuscript.

2. Two-dimensional agarose gels can be quite difficult to evaluate. The ones you show have a high level of background, but I think they are sufficient to support the conclusions you make from them. These are some of the better ones I've seen in plant organelle manuscripts, so I don't have any major concern with these in this paper. I do not have any suggestions for anything different as this work adds an additional way to look at the plants.

Rev. 2:

This is a very interesting study of the RNase H1 protein in plants that functions in both mitochondrial and plastid DNA stability. The investigators describe evidence of its role in mitochondrial genome stability and, in association, plant development. The study provides compelling evidence that the RNase H1 protein in mitochondrial participates with other components, most notably OSB, in its function, and influences homologous recombination within mitochondrial repeated sequences.

The manuscript is well written and informative. However, there are a few areas where clarification would help the reader to place this work in the context of what we currently know about plant mitochondrial and plastid genome behavior. 

1. The authors carry out much of their protein targeting work in protoplasts that appear to represent leaf mesophyll cells. However, the developmental expression of the RNase H1 is not in mesophyll but in vascular tissue, meristem, reproductive/embryo and perhaps epidermis (the images appear to show expression in trichomes?). Therefore, one assumes that the process of mitochondrial DNA replication, presumably accompanying biogenesis, is likely enriched in meristem, reproductive tissue and embryo. It is not clear why there is enhanced signal in vascular tissues, but it is possible that this gene, like many others that dually target mitochondria and plastids, may play multiple roles. At very least it is important to be precise in the text to make clear that the mesophyll chloroplasts being tested for targeting are not organelles in which this protein likely resides.

2. I would argue that there is not a "canonical" dual targeting protein configuration in plants as suggested by the authors, or they should be specific in defining it. The authors indicate that there is an internal regulatory signal that defines plastid and mitochondrial targeting behavior in response to cellular signaling. This appears to be a reasonable interpretation of the data. But this mechanism is likely not new, and the authors should have provided more information about the sequence located at 69-77 aa downstream from the translation start. Twin presequences have been described in many plant genes, and alternative translation initiation, including non-ATG start sites, have been demonstrated in a number of plant genes including those organellar targeted. One straightforward interpretation of the results presented is that the "plastid targeting" form AtRNH1C is actually controlled by a twin presequence that permits alternative translation initiation, with the domain 69-77 serving as the mitochondrial targeting domain. Translation initiation at the predicted ATG start may compete with a second translation initiation site downsteam, providing the ability of a single gene to encode two forms of the product. This type of regulation has been reported for the organellar DNA polymerase (doi: 10.1105/tpc.108.063644) and would not be truly novel, but it is a very interesting feature and the amino acid sequence in the 69-77aa region, with its likely facilitation of mitochondrial targeting, should be addressed more clearly in this report.

3. Is there evidence that the plastid form of the RNase H functions to stabilize the plastid genome, or functions within the plastid genome in the same manner as in the mitochondrion? Disruption of the mitochondrial gene is compensated by retargeting of the plastid form, but what is the consequence of disruption of the gene encoding the plastid form?

Minor issues:

In some of the figures using microscopy, arrows to indicate specific features like cristae would probably be helpful to general readers.

In Figure 3, it would be helpful to use more specific genetic description. What is the population size, what is the precise segregation?

Rev. 3:

In this manuscript, the authors studied the role of RNase H1 in the mitochondria in Arabidopsis thaliana. They argue that whilst AtRNH1B is the main RNase H1 isoform in the mitochondria, AtRNH1C can be targeted to the mitochondria when AtRNH1B is missing and compensate for its absence. A small motif in AtRNH1C prevents its targeting to the mitochondria in the presence of AtRNH1B but it is not clear how lack of AtRNH1B would relieve this self-inhibition. When both AtRNH1B&C are deficient, the plants are sterile, R-loops accumulate in the mitochondria and the frequency of HR events increases. The authors present evidence that RNH1 acts together with the ssDNA-binding protein Osb1 to control HR frequency but the mechanisms involved are unclear. Finally, the authors argue that, contrary to what happens in mammalian cells, R-loops do not participate in the replication of mitochondrial DNA but the evidence presented in support of this conclusion is rather weak. Whilst some of these observations are interesting, they fail to clarify the role of RNase H1 or R-loops in mitochondrial function or in the regulation of HR. As they stand, these observations are a bit disparate and probably not of sufficient general interest. Perhaps this study would be better suited for a more specialized journal. 

Major points: 

- Fig 1A: The distribution pattern of AtRNH1B-GFP and MitoTracker are very different and it is very difficult to conclude from these images that their infrequent colocalization does not simply happen by chance. At best, AtRNH1B-GFP would localize in a very small subset of mitochondria. The authors should (i) tone down their conclusion that AtRNH1B-GFP localizes to the mitochondria and (ii) speculate as to why they can only detect AtRNH1B-GFP in only a subset of mitochondria (what's special about those mitochondria?). 

- Fig 1C: it is not clear how the authors have determined that the upper band corresponds to the unprocessed, MTS-containing AtRNH1B? How did they exclude that this faint band could correspond to another type of isoform (phospho-isoform for example)?

- Fig 1D: to make the comparison meaningful, the authors should show clearly that the same amount of proteins has been put everywhere.

- Fig 3A: AtRNH1Bpro: AtRNH1B△MTS-GFP atrnh1b 1c+/-; is there partial complementation here? The seeds look less damaged than in atrnh1b 1c+/-. Is it because the siliques have been photographed at an earlier stage of desiccation or is it that there is partial complementation? 

- Fig 4A: please provide the magnification for Col-0 to allow a proper comparison. 

- Lane 182: "the signals were exclusively localized in mitochondria". Same comment as for Fig 1A: this feels strongly like an over-statement. 

- Lane 233: please explain exactly how the RNAi plants have been generated (detailed explanations in the methods section should be mandatory). 

- Lane 242: To say that loss of mitochondrial RNase H1 triggers sterility is different from saying that it is important for embryogenesis. On Fig S5E, it looks as if the siliques of AtRNH1BRNAi atrnh1c have not been fertilized at all, hinting at a fertilization problem distinct from the problems of embryo development suggested on Figure 3C. Considering that the mature pollen expresses a lot of AtRNH1B (Figure 2B), could it be that the plants are male or female sterile? This is easy to test. It would also help to know how the plants have been pollinated (self-pollination?) and which cells express the RNAi construct: this should be made clear in the text (see above comment). 

- Fig 7A and Fig 9: please provide the raw data (% IP) of the DRIP-qPCR in order to give an idea of the actual strength of the DRIP signal. Please also provide at least one negative locus for comparison.

- Lane 272: "These results suggested that R-loops promote HR in Arabidopsis mitochondrial genome". Here the authors show that both R-loop and HR levels increase concomitantly but it is not enough to demonstrate that R-loops promote HR. It is only a correlation. This sentence is therefore an over-statement.

- Figure 7DE: The authors should investigate using Mitotracker whether there is an overall reduction in the number of mitochondria in atrnh1b/c. Could these data be consistent with a problem of mitochondria replication? 

- Fig S6B/C: A statistical analysis of the significance of the data should be performed. 

- The results presented on Fig 8 are not very convincing. It is not clear how many times they have been performed. Quantifications should be provided to confirm the conclusions put forward by the authors. 

- Fig 9A: To validate the supposed increase in R-loop formation in osb1, it is essential to show that the increase in DRIP signal is actually sensitive to the addition of exogenous RNase H. It is unclear why the authors did not look at the same sites in Fig 7A and in Fig 9A. The putative increase in R-loop levels in the absence of Osb1 should be directly compared to the increase observed in the absence of mitochondrial RNase H1. 

- Fig 9D: The asymmetry of the HR reaction at L and I (L1/2 versus L2/1; I1/2 versus I2/1) is exactly the opposite in osb1 and in atrnh1b/c (see Fig 7C); and whilst the impact of atrnh1b/c was "symmetrical" at EE, it is strongly asymmetric in osb1. In addition, the impact of osb1 is much greater (*600) at EE2/1 than in atrnh1b/c (*30) and only partially compensated by the over-expression of AtRNH1B. At I, osb1 significantly impacts HR frequency whilst the authors did not detect an accumulation of R-loops. Taken together these observations strongly suggest that Osb1 and AtRNH1B/C do not work in the same pathway to control HR and that at the very least, a large part of the contribution of Osb1 to HR control is R-loop independent. The authors should therefore tone down their conclusions. 

- Fig 10B: is there a mistake in the legend here? The authors say: (lane 328) "AtRNH1BRNAi atrnh1c showed an epistatic effect to osb1 at repeat L and an additive effect at repeat EE", whilst the DRIP signal in osb1 RNAi #1 (blue) is systematically much lower than in either single mutant. The conclusion clearly does not match the data. 

Minor points:

- Fig 1A: to facilitate the reading of the figure, it would be better not to include the Bright field in the merge. 

- Fig S1A: The pictures are incredibly pixelated and are not of sufficient quality to be published in their current state. Explain also to the non-specialists that chloroplasts can be observed in the bright field channel as round objects. 

- Fig S1B: please explain the meaning of the different colors/domains (catalytic, HBD).

- Lane 154: "As the mitochondrial localization of AtRNH1B is responsible for embryo lethality". Should it not be: "As lack of the mitochondrial localization…" ?

- Lane 274: "Because frequent ectopic HR is a source of genome integrity". Presumably, replace "integrity" by "instability". 

- Fig 7F would probably benefit from being moved to Fig 1 to confirm the mitochondrial localization of AtRNH1B. Similarly, Fig 7G should be moved to Fig 4 to validate the mitochondrial localization of AtRNH1C in the absence of AtRNH1B. 

- Lane 313: "we assessed mitochondrial R-loops in msh1, osb1, and recA3 by DRIP-qPCR (Figure S7)". No DRIP experiments are displayed on Fig S7.

---

## [Decision Letter · Decision Letter 2]

24 Jun 2021

Dear Dr Sun,

Thank you for submitting your revised Research Article entitled "A facultative dual-targeting mechanism maintains mitochondrial RNase H1 and R-loop homeostasis for genome integrity and early embryogenesis in Arabidopsis" for publication in PLOS Biology. I have now obtained advice from two of the original reviewers and have discussed their comments with the Academic Editor. 

Based on the reviews (attached below), we will probably accept this manuscript for publication, provided you satisfactorily address the remaining points raised by Reviewer 3. Please also make sure to address the data and other policy-related requests indicated below.

In addition, we would like you to consider a suggestion to improve the title:

"Mitochondrial RNase H1 activity regulates R-loop homeostasis to maintain genome integrity and enable early embryogenesis in Arabidopsis"

We expect to receive your revised manuscript within two weeks. 

*Published Peer Review History*

*Early Version*

Sincerely,

Ines Alvarez-Garcia, PhD

Senior Editor

PLOS Biology

on behalf of

Richard Hodge,

Associate Editor,

rhodge@plos.org,

PLOS Biology

Fig 3B; Fig. 7A, C-G; Fig. 9A, C, D; Fig. 10B, C; Fig. S3B; Fig. S5B and Fig. S6B, C

We require the original, uncropped and minimally adjusted images supporting all blot and gel results reported in an article's figures or Supporting Information files. We will require these files before a manuscript can be accepted so please prepare and upload them now. Please carefully read our guidelines for how to prepare and upload this data: https://journals.plos.org/plosbiology/s/figures#loc-blot-and-gel-reporting-requirements

BLURB

Please also provide a blurb which (if accepted) will be included in our weekly and monthly Electronic Table of Contents, sent out to readers of PLOS Biology, and may be used to promote your article in social media. The blurb should be about 30-40 words long and is subject to editorial changes. It should, without exaggeration, entice people to read your manuscript. It should not be redundant with the title and should not contain acronyms or abbreviations. For examples, view our author guidelines: https://journals.plos.org/plosbiology/s/revising-your-manuscript#loc-blurb

Reviewers' comments

Reviewer #2:

I believe that the reviewers have satisfactorily addressed the concerns expressed in the previous review of this manuscript.

Reviewer #3: The authors have made substantial corrections to their manuscript and its quality has improved as a result. I still think that this story would fit better in a more specialized journal and that the role of RNase H1 in the maintenance and replication of Arabidopsis mitochondria is still not entirely clear, as admitted by the authors themselves. 

I suggest that the raw DRIP data should be shown in the supplementary material. 

I suggest that the respFigure 11 should be added to the manuscript to balance the conclusions that RNase H1 does not play a role in the replication of mitochondria in Arabidopsis. 

Corrections to the text: 

L226: replace "increasement" by "increase". 

L228: "there are much more immunogold labeled GFP particles detected in atrnh1b chloroplast than that in atrnh1b chloroplast" should be "than in atrnh1c chloroplasts".

---

## [Editor Report · Decision Letter 3]

8 Jul 2021

Dear Dr Sun,

On behalf of my colleagues and the Academic Editor, Xuemei Chen, I am pleased to say that we can in principle offer to publish your Research Article "Mitochondrial RNase H1 activity regulates R-loop homeostasis to maintain genome integrity and enable early embryogenesis in Arabidopsis" in PLOS Biology, provided you address any remaining formatting and reporting issues. These will be detailed in an email that will follow this letter and that you will usually receive within 2-3 business days, during which time no action is required from you. Please note that we will not be able to formally accept your manuscript and schedule it for publication until you have made the required changes.

PRESS

Sincerely, 

Richard

Richard Hodge, PhD

Associate Editor, PLOS Biology

rhodge@plos.org

PLOS
